# Unusual KIE and dynamics effects in the Fe-catalyzed hetero-Diels-Alder reaction of unactivated aldehydes and dienes

Yuhong Yang[1,2,3], Xiaoyong Zhang [2,3], Li-Ping Zhong[2], Jialing Lan[1,2], Xin Li[2], Chuang-Chuang Li [2] & Lung Wa Chung [2✉]

Hetero-Diels-Alder (HDA) reaction is an important synthetic method for many natural products. An iron(III) catalyst was developed to catalyze the challenging HDA reaction of unactivated aldehydes and dienes with high selectivity. Here we report extensive density-functional theory (DFT) calculations and molecular dynamics simulations that show effects of iron (including its coordinate mode and/or spin state) on the dynamics of this reaction: considerably enhancing dynamically stepwise process, broadening entrance channel and narrowing exit channel from concerted asynchronous transition states. Also, our combined computational and experimental secondary KIE studies reveal unexpectedly large KIE values for the five-coordinate pathway even with considerable C–C bond forming, due to equilibrium isotope effect from the change in the metal coordination. Moreover, steric and electronic effects are computationally shown to dictate the C=O chemoselectivity for an α,β-unsaturated aldehyde, which is verified experimentally. Our mechanistic study may help design homogeneous, heterogeneous and biological catalysts for this challenging reaction.

[1] School of Chemistry and Chemical Engineering, Harbin Institute of Technology, Harbin 150001, China. [2] Shenzhen Grubbs Institute, Department of Chemistry and Guangdong Provincial Key Laboratory of Catalytic Chemistry, Southern University of Science and Technology, Shenzhen 518055, China. [3] These authors contributed equally: Yuhong Yang, Xiaoyong Zhang. ✉email: oscarchung@sustech.edu.cn

The Diels–Alder (DA) reaction has long been one of the most popular, classical, and atom-economical chemical and biological synthetic methods for forming six-membered carbocycles[1–6]. This important cycloaddition reaction was extended to the addition of other substrates containing C=X bonds (e.g., X=O; i.e., oxa-Diels–Alder (ODA) reaction) to give various six-membered heterocyclic molecules, which are very useful in the synthesis of carbohydrates and many natural products[1,2,7–9]. Compared with the DA reaction, an uncatalyzed ODA reaction with carbonyl compounds is synthetically more challenging (due to their stronger C=O bonds) and generally requires activated aldehydes (e.g., glyoxylates or ketomalonate) or electron-rich dienes (e.g., Danishefsky- or Rawal-type dienes)[8,9]. Alternatively, the ODA reaction with unactivated aldehydes and simple dienes can be catalyzed by a very strong Brønsted or Lewis acid to activate the aldehyde, but its poor functional-group tolerance limits the substrate scope.

Recently, a bioinspired and earth-abundant cationic iron(III) porphyrin catalyst ([Fe(TPP)]BF$_4$, where TPP is *meso*-tetraphenylporphyrinato) was developed and overcomes these synthetic difficulties to catalyze the ODA reaction of unactivated aldehydes and simple dienes under relatively mild conditions (Fig. 1)[10]. Excellent regio-, stereo-, and chemoselectivity were also observed for the ODA reaction (Fig. 1). Especially, only the rare chemoselective cycloaddition with the C=O bond (e.g., **E**) was observed even in the presence of the other unsaturated bonds (e.g., alkene, alkyne, diene, nitrile, or nitro groups), and with high functional-group tolerance (halide, ether, imide, acetal, acetoxy, or siloxy groups)[10]. Notably, the Fe(III) catalyst can undergo the ODA reaction with cyclohexenone, which is even less reactive. Thus, this vital study on bioinspired Fe(III) has inspired the design of a metal–organic framework (MOF) system to catalyze such challenging ODA reactions[11].

However, the reaction mechanism of the unique reactivity and high selectivity of the Fe(III)-catalyzed challenging ODA of unactivated aldehydes and dienes (Fig. 1) remains unclear and is much more complicated than that of the uncatalyzed ODA

reaction due to various possible spin states and coordinate modes of Fe. Also, the reaction dynamics of metal-catalyzed DA or HDA reactions, including this Fe-catalyzed ODA reaction, are completely unknown.

Computational chemistry has played a vital role in elucidating the mechanisms of organic reactions and helping design better catalysts and/or reactions[12–20]. Although many computational studies on the mechanism of DA and 1,3-dipolar cycloaddition reactions have been reported[3,4,6,21–26], computational works on the mechanism of HDA reactions are quite limited[27–29]. In addition, Carpenter, Hase, Houk, Singleton, Tantillo, and other groups carried out quasi-classical molecular dynamics (e.g., MD with a density-functional theory (DFT) method) studies to provide several new mechanistic concepts for several uncatalyzed DA reactions and other (almost metal-free) organic and enzymatic reactions (e.g., dynamically concerted and dynamically stepwise mechanisms, energy-dependent reaction selectivity, and dynamically controlled selectivity)[30–43]. These DFT MD studies offered us time-resolved mechanistic insights and helped elucidate the timing of the bond formation in (bio)chemical reactions.

To elucidate the reaction mechanism and reaction dynamics, as well as to continue our studies on bioinspired catalysis and dynamics[44–49], we carried out extensive DFT calculations and DFT quasi-classical MD simulations, as well as two mechanistic experiments on the Fe(III)-catalyzed ODA reaction (Fig. 1). This study is also the first DFT quasi-classical MD simulation on Fe-catalyzed organic reactions.

Herein, our DFT calculations and MD simulations show pronounced effects of iron (including its coordinate mode and/or spin state) on the dynamics of this ODA reaction (some dynamically stepwise process, broader entrance channel, and narrower exit channel from concerted asynchronous transition states). In addition, our computational and experimental secondary KIE studies reveal large KIE values for the five-coordinate pathway even with significant C–C bond formation. Moreover, steric and electronic effects are computationally found to dictate the C=O chemoselectivity for an α,β-unsaturated aldehyde, which is verified experimentally.

## Results

**DFT calculations on the reaction mechanism.** The SMD B3LYP-D3//B3LYP-D3 method with a mixed basis set (6-31G* for the C, H, O, and N atoms, def2-TZVP for the Fe atom) and a Fe(III)–porphine model catalyst were mainly used to study the reaction mechanism in this study unless otherwise stated. The key results on the mechanism of the uncatalyzed and Fe(III)-catalyzed reactions, as well as the DFT quasi-classical MD results, are presented in the main text. The other detailed results and discussion (including the effect of oriented external electric field (OEEF)[3,50–53], the origin of its high regio- and stereoselectivity, as well as the less favorable pathways) are provided in the Supplementary Information.

**The formation of product A.** The reaction of benzaldehyde (PhCHO) and 2,3-dimethyl-1,3-butadiene forms cycloaddition product **A** (Fig. 1). Our SMD B3LYP-D3//B3LYP-D3 and SMD B3LYP-D3 calculations indicate that the uncatalyzed *endo*-ODA reaction requires a very high barrier of 36.0–37.3 kcal mol$^{-1}$ and follows a concerted synchronous mechanism in a singlet state (Fig. 2). In addition, comparable barriers were obtained by the SMD M06-D3//M06-D3 (34.7 kcal mol$^{-1}$) and SMD M06-2X-D3//M06-2X-D3 (38.4 kcal mol$^{-1}$) methods (Supplementary Table 1). These computational results support that the ODA reaction cannot proceed without a catalyst under these reaction conditions due to the high barrier.

**Fig. 1 Representative ODA reactions catalyzed by the Fe(III) catalyst.** Selected Fe-catalyzed ODA products with their yields and selectivities.

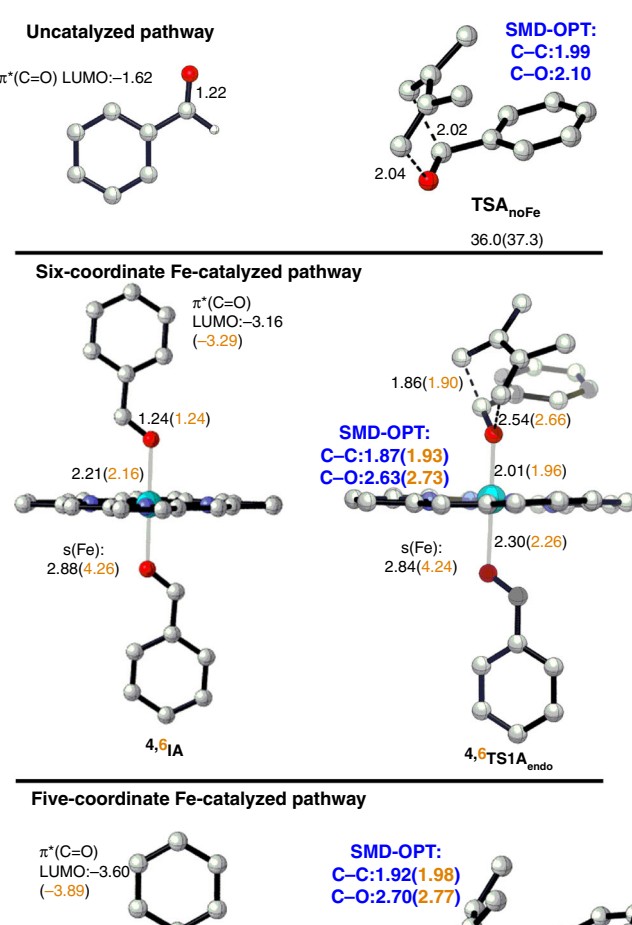

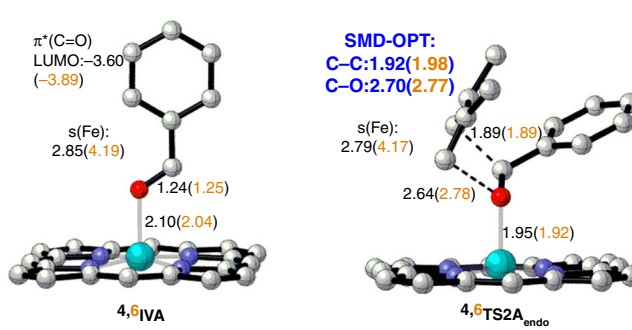

**Fig. 2 Key structures for the uncatalyzed and Fe-catalyzed ODA reactions.** Optimized key structures for the formation of **A** in the singlet state for the uncatalyzed pathway (top) as well as in the quartet and sextet states (in orange) for the (middle) six-coordinate and (bottom) five-coordinate Fe-catalyzed pathways are shown. The computed reaction barriers and reaction energies for the uncatalyzed reaction by the SMD B3LYP-D3//B3LYP-D3 and SMD B3LYP-D3 (in parentheses) methods, key distances (Å, in italics, with the two key bonds optimized in solvent highlighted in blue), LUMO energy (eV), and spin density (s) on Fe by the SMD B3LYP-D3//B3LYP-D3 method are given. Unimportant hydrogen atoms are not shown for clarity.

In sharp contrast, our DFT results showed that the iron catalyst significantly impacted the electronic structures, the energetic profile, as well as the mechanism of the ODA reaction. As shown in Fig. 3, the coordination of two PhCHO molecules (as axial ligands) to the $d^5$ Fe(III) metal[10] in the quartet state (six-coordinate mode, **4IA**) is the most stable species before the cycloaddition reaction. For instance, **4IA** is lower in free energy than **6IA** and **2IA** by ~6.2 and 10.2 kcal mol$^{-1}$, respectively, by the SMD B3LYP-D3//B3LYP-D3 method. The computed free-energy difference between **4IA** and **6IA** can be reduced to 1.3–4.8 kcal mol$^{-1}$ by the SMD PBE0-D3//B3LYP-D3 and SMD B3PW91-D3//B3LYP-D3 methods (Supplementary Table 3A). Since several Fe(III)–porphyrin complexes with two axial oxygen

ligands were experimentally found to have admixed spin states (quartet and sextet states)[54], the B3LYP-D3 method modestly overestimates the stability of **4IA** relative to **6IA**. In addition, **4IA** is more stable than the five-coordinate intermediate **4IVA** with one axial PhCHO ligand by 4.1–4.8 kcal mol$^{-1}$ by the SMD B3LYP-D3//B3LYP-D3 and SMD B3LYP-D3 methods.

When PhCHO coordinates to the Lewis-acidic Fe(III) metal in **4IA** or **4IVA**, the C=O bond of PhCHO was elongated by ~0.02 Å and, especially, such coordination to the metal significantly lowered the energy of an unoccupied orbital for the $\pi^*_{(C=O)}$ moiety (free PhCHO: −1.62 eV; **4,6IA**: −3.16~−3.29 eV; particularly, **4,6IVA**: −3.60~−3.89 eV, see Fig. 2). Such carbonyl LUMO-lowering activation through the Fe–O interaction promotes the reaction with an electron-rich diene. The decreased Pauli repulsion between the reactants has recently been suggested to also accelerate the Lewis-acid-catalyzed Diels–Alder reactions[22]. Interestingly, compared with **4IA** or **4IVA**, **6IA** and **6IVA** have a shorter Fe–O bond(s) by 0.05–0.06 Å and, thus, lower $\pi^*_{(C=O)}$ orbital energies as well as lower intrinsic barriers (see the Discussion below).

Moreover, our SMD B3LYP-D3//B3LYP-D3 and SMD B3LYP-D3 results further elucidate that two concerted *endo*-cycloaddition pathways (through either six- or five-coordinate mode pathway) in the quartet state are the most favorable pathways and have the lowest barriers (20.3–24.5 kcal mol$^{-1}$, Fig. 3; the *exo*-ODA reaction has higher barriers, see Supplementary Fig. 2), which are much lower than that of the uncatalyzed reaction (by about 15.0–17.9 kcal mol$^{-1}$, Fig. 2). In addition, the six-coordinate mode pathway via **4TS1A_endo** has the lowest reaction barrier, which is just lower by ~0.3 kcal mol$^{-1}$ than that for the five-coordinate mode pathway via **4TS2A_endo** by the SMD B3LYP-D3 method, due to the lower carbonyl LUMO energy in **4IVA**. Notably, driven by the above-mentioned larger carbonyl LUMO activation caused by the high-spin metal and the nature of admixed spin states[54], the reaction barriers of these two pathways in the sextet state are also slightly higher than those of their quartet counterparts by 0.3–1.4 kcal mol$^{-1}$ by the SMD B3LYP-D3 method. Finally, the six-coordinate mode pathway to form product **4IIIA_endo** is exothermic by 2.6 kcal mol$^{-1}$. Whereas the five-coordinate mode pathway to form product **4VIA_endo**, which is slightly endergonic by 0.4 kcal mol$^{-1}$ relative to **4IA**, might be a reversible process. Another PhCHO substrate should coordinate to **4VIA_endo** to form a more stable six-coordinate mode product **4IIIA_endo**. The relative energy of the dissociation of the product **A** along with re-coordination of another aldehyde molecule to regenerate **4IA** is about 0.5 kcal mol$^{-1}$ (Fig. 3 and Supplementary Table 3B). Importantly, these results, which are also qualitatively supported by different DFT functionals (the B3PW91-D3, PBE0-D3, ωB97XD, PW6B95-D3, OLYP-D3, and OPBE-D3 methods, see Supplementary Table 2), suggest that these two coordinate modes in two different spin states (quartet and sextet) can contribute to the ODA reaction, representing two-mode reactivity with admixed spin states[55].

The state-of-the-art DLPNO-CCSD(T) method was also employed to evaluate the relative energy of the critical structures and the related Fe(III) complex coordinating with two acetones as axial ligands[54] (see Supplementary Tables 2 and 17 as well as Supplementary Fig. 10). However, compared with the observed admixture states[54], this method was found to significantly overstabilize the high-spin states[56] than the quartet state ($\Delta G_{S-Q}$ = ~−11.8 kcal mol$^{-1}$). Whereas the PBE0 method was a good method to describe this energy gap[57] and also supported the SMD B3LYP-D3 results in this system.

Interestingly, dispersion attraction[58] by the SMD B3LYP-D3 method was found to lower these reaction barriers and enhance

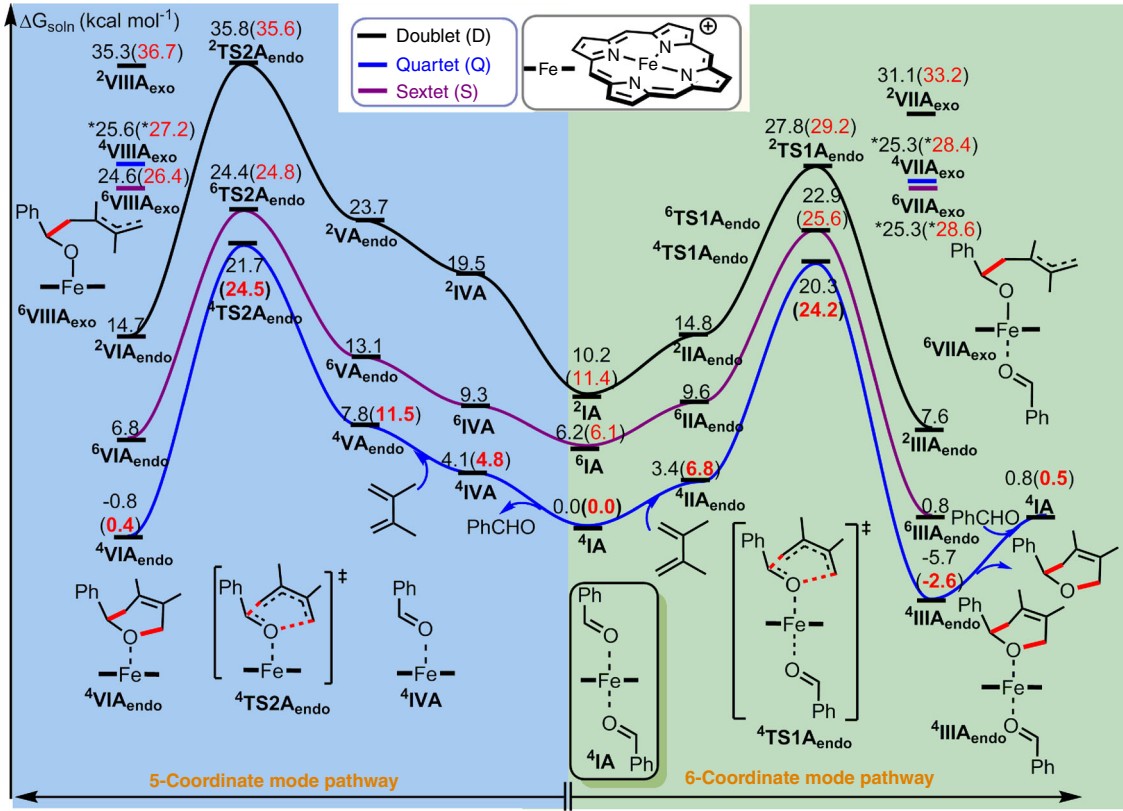

**Fig. 3 Free-energy profile of the Fe-catalyzed ODA reaction.** The key pathways to form product **A** in three spin states in solution by the SMD B3LYP-D3//B3LYP-D3 and SMD B3LYP-D3 (in parentheses) methods are given.

the driving force of the reaction by roughly ~8–17 kcal mol$^{-1}$, compared with those structures optimized by the SMD B3LYP method (excluding the dispersion effect). For instance, the reaction barrier without dispersion via $^4$**TS1A**$_{endo}$ becomes ~41.1 kcal mol$^{-1}$ (Supplementary Table 9). Furthermore, the two lowest-energy stepwise intermediates involving one carbon–carbon bond formation ($^6$**VIIA**$_{exo}$ and $^6$**VIIIA**$_{exo}$, Fig. 3) were found to be higher in free energy than the lowest-energy concerted transition state, $^4$**TS1A**$_{endo}$, by at least 2.2–4.4 kcal mol$^{-1}$ by the SMD B3LYP-D3 method. Overall, these computational results support the concerted mechanism as the primary pathway.

In addition, the new C–C bond distance in $^{4,6}$**TS1A**$_{endo}$ and $^{4,6}$**TS2A**$_{endo}$ was found to be shorter (~1.86–1.90 Å, Fig. 2) than that in the uncatalyzed reaction (2.02 Å), whereas the new C–O bond distances were much longer (~2.54–2.78 Å) than that in the uncatalyzed reaction (2.04 Å). Strikingly, the bond between the Fe and the reacting carbonyl O was also shortened in $^{4,6}$**TS1A**$_{endo}$ (by ~0.20 Å) and $^{4,6}$**TS2A**$_{endo}$ (by ~0.12–0.15 Å) compared with that in $^{4,6}$**IA** and $^{4,6}$**IVA**. Such bond strengthening is analogous to the enhanced hydrogen bonds in DA or ODA transition states observed in previous works[31,59–61]. Moreover, our Mulliken charge analysis further suggested that the porphyrin ligand acts as an auxiliary electron reservoir during the ODA reaction (see Supplementary Tables 5-7). In contrast to the above-mentioned enhanced Fe–O bonding, the bond between the Fe and nonreacting carbonyl O atoms becomes elongated in $^{4,6}$**TS1A**$_{endo}$ (by 0.09–0.10 Å), and the Fe–N bonds became marginally longer in $^{4,6}$**TS1A**$_{endo}$ and $^{4,6}$**TS2A**$_{endo}$ (by 0.01–0.02 Å). Therefore, the iron catalyst not only reduces the reaction barrier, but also makes the reaction follow a concerted asynchronous mechanism in the quartet or sextet state, and adapts different coordination/bonding configurations to further activate the carbonyl substrate.

**DFT and experimental secondary KIE studies.** Kinetic isotope effect (KIE) has been widely employed as a diagnostic tool to help determine the mechanism of many chemical and biochemical reactions, as well as probe the transition-state structures involved[23–26,43,62,63]. Secondary deuterium ($k_H/k_D$) KIE studies for DA or ODA reactions with RCHO/RCDO should give a smaller KIE value for a later transition state (with a shorter C–C bond formation leading to more $C_{sp3}$ character and less $C_{sp2}$ character)[63]. Our combined computational and experimental secondary $k_H/k_D$ KIE studies for the Fe-catalyzed ODA reaction were carried out to help determine the mechanism (Fig. 4).

A measured KIE value of 0.926 (±0.007) was determined from three runs of competitive experiments (Fig. 4b). Notably, this measured KIE value is in between the computed KIE values for the six- and five-coordinate mode pathways (0.886–0.904 and 0.950–0.968, respectively; Fig. 4a), supporting that both the six- and five-coordinate mode pathways should be involved in the Fe-catalyzed ODA reaction. Whereas the computed similar KIE values for two different possible spin states are less conclusive. Surprisingly, the computed KIE values for the six-coordinate Fe-catalyzed reaction and the uncatalyzed reaction (0.886–0.904) were similar, but they were quite different from that for the five-coordinate Fe-catalyzed reaction (0.950–0.968), even though these three transition states had comparable C–C bond distances (1.86–2.02 Å, Fig. 2). Therefore, the higher KIE values for the five-coordinate mode pathway cannot truly reflect its transition states with considerable C–C bond formation. Such higher KIE values for the five-coordinate mode pathway were attributed to its unusual and nonnegligible secondary equilibrium isotope effect (EIE) derived from the dissociation of one benzaldehyde from **IA** catalyst (~1.076, Fig. 4c). The similar KIE and EIE (~1.069–1.082) values were also obtained from the SMD B3LYP-D3, SMD

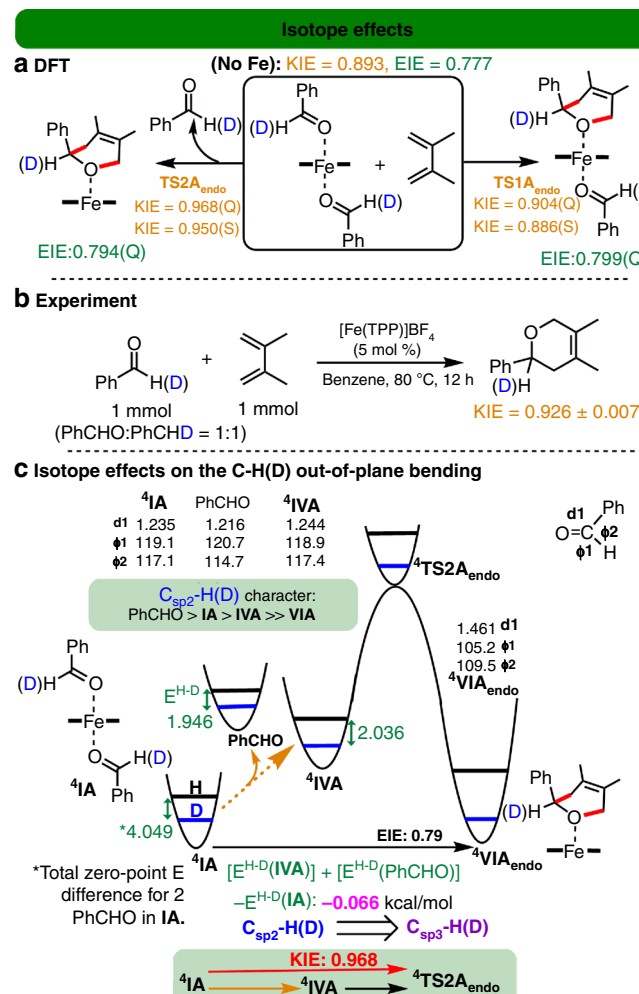

**Fig. 4 Secondary isotope effects of the Fe-catalyzed ODA reaction.**
**a** Computed secondary $k_H/k_D$ KIE and EIE results for the formation of **A** in the quartet (Q) and sextet (S) states by the B3LYP-D3 method. **b** Experimental $k_H/k_D$ KIE result for the formation of **A**. **c** Schematic of the isotope effects and the computed zero-point energy differences ($E^{H-D}$, in kcal mol$^{-1}$) of the key structures by the B3LYP-D3 method.

PBE0-D3, and SMD B3PW91-D3 methods (Supplementary Table 14). The coordination of PhCHO to the Fe(III) complex gives rise to a longer C–O bond and smaller H–C–O/H–C–C bond angles of the PhCHO group, which increases its $C_{sp3}$ character, resulting in a larger zero-point energy difference for the H/D isotopes ($E^{H-D}$, Fig. 4c)[63]. Accordingly, the measured secondary KIE value does not always fully manifest the exact transition-state structure, as EIE derived from usually unrecognized coordination changes in the metal could mask the overall KIE.

**The charge effect of the Fe complexes.** Owing to lower Lewis acidity of the Fe metal, our additional SMD B3LYP-D3 results showed that the two neutral Fe(III) complexes with coordination of one anionic Cl$^-$ or OTf$^-$ ligand in the five-coordinate mode $^{4,6}$**IVA(Cl–)** and $^{4,6}$**IVA(OTf–)** are the most stable species (without coordination of a PhCHO substrate) before ODA reaction (see Supplementary Fig. 11). $^6$**IVA(Cl–)** and $^4$**IVA (OTf–)** are just about 1.3 and 1.4 kcal mol$^{-1}$ lower in free energy than $^4$**IVA(Cl–)** and $^6$**IVA(OTf–)**, respectively. Coordination of one PhCHO to **IVA(Cl–)** or **IVA(OTf–)** to form the six-

coordinate intermediates $^4$**IA(Cl–)** or $^6$**IA(OTf–)** was computed to be endergonic by 9.5 and 3.7 kcal mol$^{-1}$, respectively. Consequently, their overall barriers for the ODA reaction become much higher (29.9–36.0 kcal mol$^{-1}$ via $^4$**TS1A$_{endo}$(Cl–)** and $^4$**TS1A$_{endo}$(OTf–)**) than the cationic Fe complex $^4$**IA** via $^4$**TS1A$_{endo}$**. These results qualitatively explain no observed reactivity of these two neutral Fe complexes.

In addition, the effects of the ruthenium metal and biological ligand were also examined (see Supplementary Figs. 6–9 and Supplementary Table 16). A considerable Ru(II) character in the ground-state complex $^2$**IA$_{Ru}$** and a comparable reactivity using one axial histidine ligand were found.

**DFT and experimental studies on the chemoselectivity.** One of the most salient features in this Fe(III)-catalyzed ODA reaction is the chemospecific cycloaddition to the inert C=O bond in the presence of the other unsaturated bonds (such as C=C bonds, which form product **E**, see Fig. 1)[10]. However, our DFT study showed that the uncatalyzed cycloaddition preferentially occurred at the C=C bond of two simple α,β-unsaturated aldehydes (3-methyl-2-butenal and 2-propenal), and their barriers to form **E1C** and **E2C** (cf. Fig. 5) were much lower than the barriers for the addition to the C=O bond by at least 4.3 kcal mol$^{-1}$. Notably, the Fe(III) catalyst reverses and significantly favors the chemoselective cycloaddition to the strong C=O bond of 3-methyl-2-butenal, forming the desired and unusual product **E1O** with a lower barrier than addition to the C=C bond by 3.5 kcal mol$^{-1}$ by the SMD B3LYP-D3 method (Fig. 5a (top)). However, our DFT calculations further predicted that the barrier to form common product **E2C** via the cycloaddition to the C=C bond of 2-propenal is lower than that to the C=O bond by ~2.4 kcal mol$^{-1}$ by the SMD B3LYP-D3 method (Fig. 5a (bottom)). One additional experiment on the reaction of 2-propenal with 2-phenyl-1,3-pentadiene was performed to verify our DFT prediction (Fig. 5b). Pleasingly, common cycloaddition product **E2C** was obtained in 62% yield, but **E2O** was not observed. These combined computational and experimental results demonstrated that the Fe and the substituents at the β-position (which increase its steric bulkiness and electron density, disfavoring the reaction with the C=C bond) play vital roles in reversing the chemoselective cycloaddition for the strong C=O bond.

**Quasi-classical MD simulations for the formation of A.** Recently, the Houk group reported seminal DFT quasi-classical MD studies on uncatalyzed DA reactions, and suggested a new mechanistic concept: dynamically concerted and dynamically stepwise mechanisms[30–32]. They proposed that a trajectory with the time gap of the two bond formation (C–C bond <1.6 Å) of less than 60 fs was considered as dynamically concerted; otherwise a trajectory was categorized as dynamically stepwise[30]. His group also showed that increasing the temperature (1180 K) or introducing polar functional groups can slightly raise the probability of the dynamically stepwise mechanism by ~2–3%. Inspired by these seminal DFT MD studies, we also conducted extensive DFT quasi-classical MD simulations (in total, 1310 trajectories conducted by using Progdyn code[64], see Table 1) on the uncatalyzed and Fe-catalyzed endo-ODA reactions to form product **A** in the gas and solution phases. Generally, the overall reaction dynamics in the gas and solution phases are qualitatively the same, but the timing of the key bond formation in the Fe-catalyzed reaction can be changed by the solvent to some extent (see the discussion below). As shown in Fig. 6 and 7 as well as Table 1, Supplementary Figs. 25–41, and Supplementary Movies 1–4, compared with the uncatalyzed reaction, several new dynamical features were observed for the Fe-catalyzed ODA reactions.

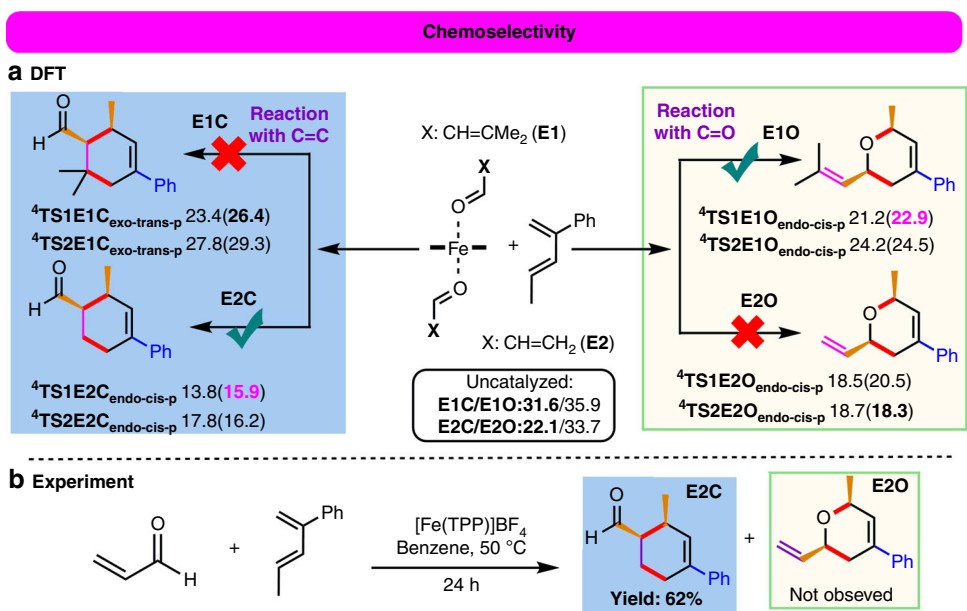

**Fig. 5 Chemoselectivity in the formation of product E. a** Free-energy profile (in kcal mol$^{-1}$) for the lowest-energy pathways to give **E1O**-type and **E2C**-type products by the SMD B3LYP-D3//B3LYP-D3 and SMD B3LYP-D3 (in parentheses) methods. **b** Experimental results for the reaction with 2-propenal.

**Table 1 Number of the DFT quasi-classical trajectories.**

|  | Gas | Soln | Soln + OEEF |
|---|---|---|---|
| Uncatalyzed | 100(0$^c$)[5$^d$] | 100(0$^c$)[6$^d$] | -- |
| 6-c $^4$Fe$^a$ | 100(21$^c$)[40$^d$] | 130(26$^c$)[50$^d$] | 130(86$^c$)[174$^d$] |
| 5-c $^4$Fe$^b$ | 130(39$^c$)[59$^d$] | 130(54$^c$)[98$^d$] | 130(88$^c$)[235$^d$] |
| 5-c $^6$Fe$^b$ | 100(74$^c$)[103$^d$] | 130(87$^c$)[199$^d$] | 130(96$^c$)[414$^d$] |

Our simulations include the uncatalyzed and Fe-catalyzed *endo*-ODA reaction in the gas phase, solution (Soln), and solution in the presence of an OEEF (with a strength of −0.003 au).
$^a$6-c stands for the six-coordinate mode pathway.
$^b$5-c stands for the five-coordinate mode pathway.
$^c$Percentage (%) of the dynamics stepwise trajectories are given in parenthesis.
$^d$Time gap (fs) of the two bond formation averaged over the productive trajectories (those which do not form the C–O bond after 900 fs is not considered).

First, the productive trajectories for the uncatalyzed reaction apparently follow synchronous dynamics, whereas the trajectories for the Fe-catalyzed reactions show asynchronous dynamics with the considerable C–C bond formation prior to the C–O bond formation. In addition, the dynamics of the Fe-catalyzed reaction have a broader entrance channel (the key C–C forming process) and a narrower exit channel (the key C–O forming process) with respect to the dynamics of the uncatalyzed reaction (Fig. 6). Moreover, an ultrafast product-forming process (mean: ~32–34 fs) as well as a very small time gap between the C–C and C–O bond formation (mean: ~5–6 fs) were generally found in the gas- and solution-phase trajectories propagated from the uncatalyzed transition-state region (Fig. 7). This very small average time gap in the ODA reaction is similar to those previously reported to symmetrical DA reactions[30]. Therefore, the uncatalyzed ODA reaction should clearly be categorized as both energetically and dynamically concerted. In addition, for the uncatalyzed ODA reaction in the gas and solution phases, ~59–79% of the trajectories form the C–C bond first with a time gap of up to ~17–18 fs, while ~19–37% of the trajectories favor the formation of the C–O bond first with a time gap of below ~8–19 fs (Fig. 7). About 2–4% of the trajectories form the C–C and C–O bonds simultaneously.

Comparatively, as discussed previously and shown by minimum energy paths (MEPs) in Fig. 6, the Fe-catalyzed reactions

involve concerted asynchronous transition states (in order of increasing asynchronicity: $^4$TS1A$_{endo}$ < $^4$TS2A$_{endo}$ < $^6$TS2A$_{endo}$, see Fig. 2). Accordingly, all the Fe-catalyzed trajectories begin with C–C bond formation, and the C–O bond formation generally proceeds after crossing the transition-state (TS) region. Also, their average time to form the product and the average time gap were found to be much longer than those of the uncatalyzed reaction. For example, the average product formation time and the average time gap for the six-coordinate quartet trajectories are increased to about ~86–103 and ~40–50 fs, respectively. Notably, dynamically concerted pathway is reduced to ~74–79% of the trajectories, while ~21–26% of the trajectories follow the dynamically stepwise pathway for this Fe-catalyzed system (Table 1). Moreover, the average product formation time and the average time gap were found to be further increased for the five-coordinate quartet trajectories (~123–149 and 59–98 fs, respectively) and particularly for the five-coordinate sextet trajectories (~182–240 and 103–199 fs, respectively). As a result, the probability of following a dynamically stepwise pathway increased to ~39–54 and ~74–87% for the five-coordinate quartet and sextet cases (Table 1), respectively. In addition, the average time gap can be further increased from 40 to 103 fs in the gas phase to 50–99 fs in the solution phase for the Fe-catalyzed reactions, but the time gap for the uncatalyzed reaction is almost unchanged in solution.

Furthermore, the average time gap for the Fe-catalyzed reactions in solution was predicted to be further extended in the presence of an OEEF along the reactive Fe–O bond axis with a strength of −0.0030 au (~174 fs for the six-coordinate system and at least ~235–414 fs for the five-coordinate systems). Notably, the C–O bond forming is still longer than 1.6 Å in some of the five-coordinate sextet trajectories with an OEEF after 900-fs simulations. Furthermore, inclusion of an OEEF significantly favors these three Fe systems being dynamically stepwise more (86–96%, see Table 1). In the dynamically stepwise trajectories for the Fe-catalyzed reactions, a few full C–C bond vibrations can occur after passing through the transition-state region before complete C–O bond formation (Supplementary Figs. 28 and 31–39). In particular, in the presence of an OEEF, one out of the 50 productive trajectories for the quartet five-coordinate mode

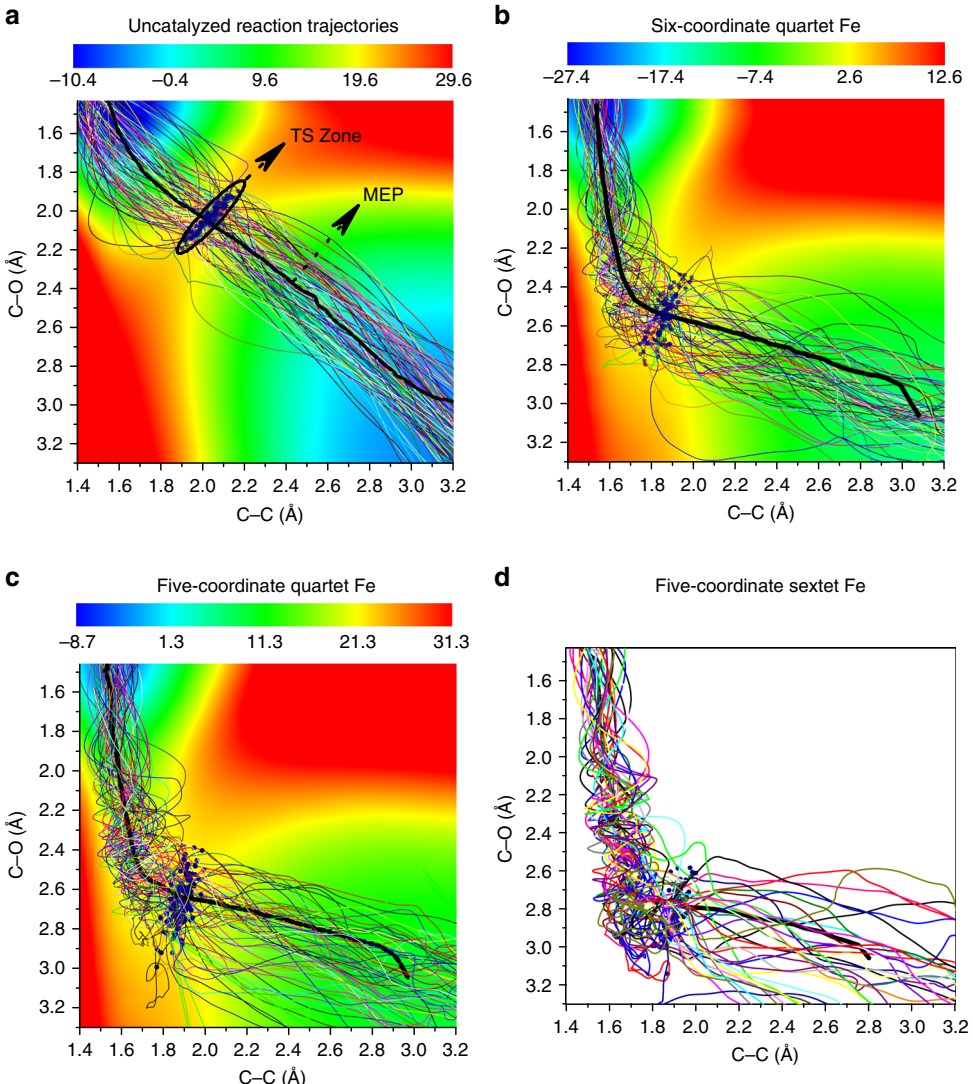

**Fig. 6 Gas-phase productive trajectories for the formation of product A.** Trajectories for **a** the uncatalyzed reaction and the Fe-catalyzed reaction of **b** the six-coordinate and quartet state, **c** the five-coordinate and quartet state, and **d** the five-coordinate and sextet state. The contour plots (energy in units of kcal mol$^{-1}$) were computed with respect to the isolated reactants (it should be noted that the color bars of the potential energy surfaces have different scales). The minimum energy path (MEP) is shown in bold, and transition-state (TS) zone is defined as a zone comprising those sampled TS structures. Throughout those Fe-catalyzed trajectories, the spin density of Fe was found to have only small changes (Supplementary Fig. 27).

pathway, and 16 out of the 34 productive trajectories for the sextet five-coordinate mode pathway, do not have the C–O bond formation and stay at the intermediate region (where C–C bond is formed, but C–O bond length is still larger than 2.2 Å) after 900 fs propagated from the TS region. This region has some features indicative of a "dynamic intermediate", as previously proposed by Singleton[34]. In addition, although a negative correlation between the two Fe–O bonds was observed for the six-coordinate quartet case in the MEPs and transition states, the bond between the Fe and the nonreacting oxygen was found to be significantly elongated at the end of some of the trajectories (Fig. 2 and Supplementary Fig. 40). These results further illustrate the subtle differences between the static and dynamic bonding characteristics.

Overall, apart from increasing the temperature or adding polar substituents[30], our DFT MD simulations suggest for the first time that even though from concerted asynchronous transition states, changes in the Fe coordination mode and/or spin state, as well as the addition of an OEEF, can significantly modulate and increase the probability of dynamically stepwise mechanism (~21–87% in

the absence of an OEEF; 86–96% in the presence of an OEEF): coordination-, spin-, or OEEF-controlled dynamics. It is worth noting that a rather long lifetime (>900 fs) of the dynamic intermediate found in some five-coordinate Fe-catalyzed trajectories in the presence of the OEEF may suggest that OEEF can increase the lifetime and may even capture the dynamic intermediate.

## Discussion

Our thorough DFT calculations and quasi-classical MD simulations provided several new mechanistic insights into the unusual effects of iron on the mechanism, secondary KIE, and dynamics in the synthetically challenging Fe-catalyzed ODA reaction of the unactivated substrates (Fig. 8). Our combined computational and experimental secondary KIE studies supported that the five- and six-coordinate mode pathways in the quartet and sextet states can be involved in the reaction (two-mode reactivity with admixed spin states) through a concerted asynchronous mechanism. Remarkably, unusually large secondary KIE values were observed and failed to correctly reflect the five-coordinate transition-state

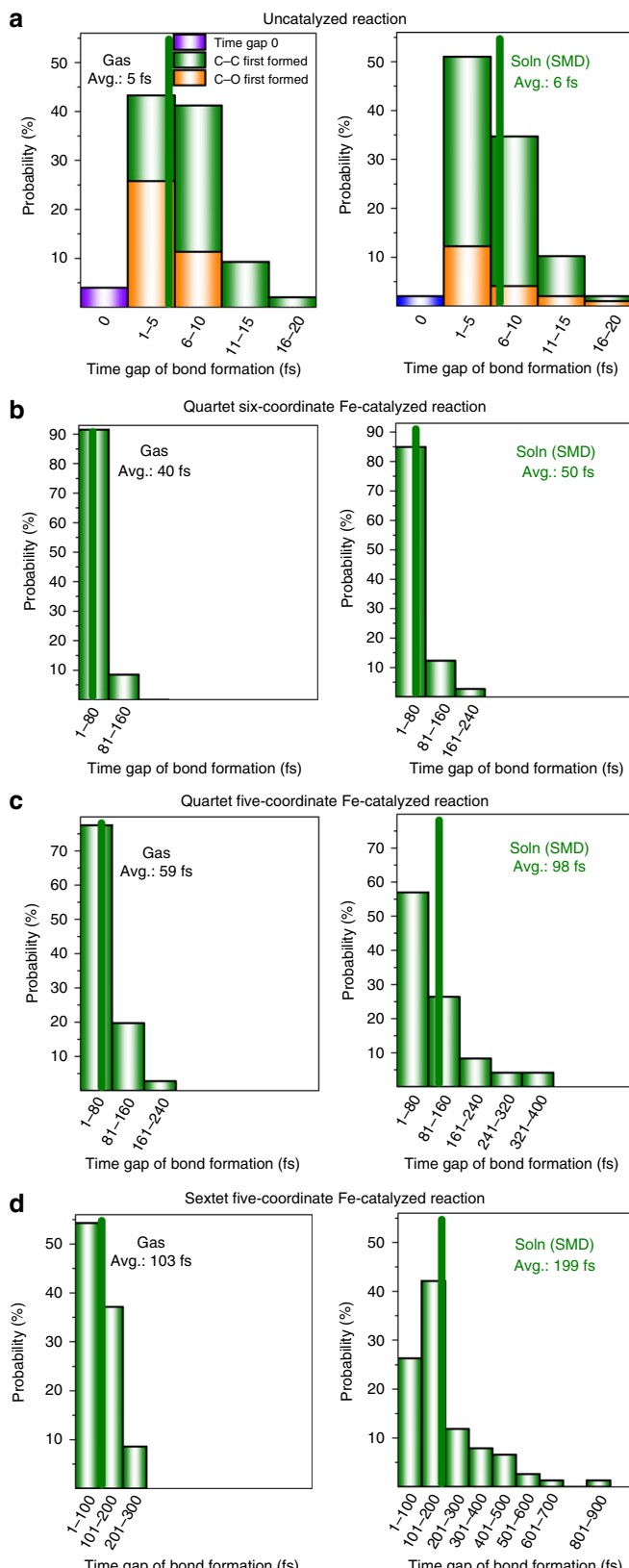

**Fig. 7 Time-gap distribution of the bond formations in the ODA reactions.** Distribution in the time gap (in fs) between C–C and C–O bond formation for **a** the uncatalyzed reaction, **b** the six-coordinate and quartet-state Fe-catalyzed reaction, **c** the five-coordinate and quartet-state Fe-catalyzed reaction, and **d** the five-coordinate and sextet-state Fe-catalyzed reaction in the gas phase (Gas, left) and solution (Soln, right).

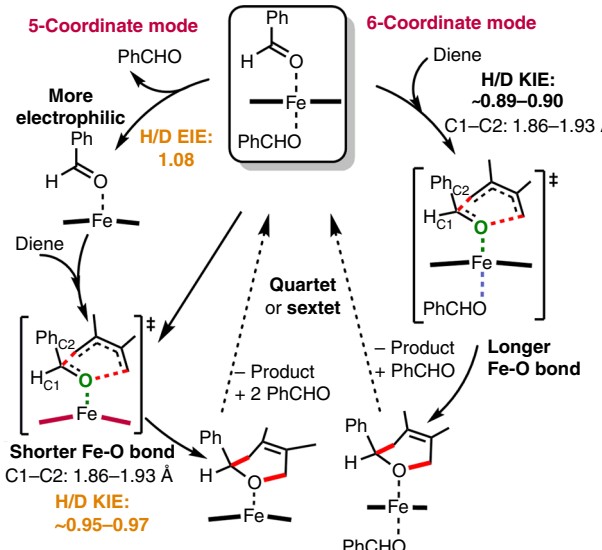

- Static: concerted asynchronous mechanism
- Two-mode reactivity with admixed spin states
- Unexpectedly high H/D KIE values for the 5-c pathway
- Dynamically-concerted and stepwise (spin-, coordination, or OEEF-controlled dynamics)
- Broader entrance and narrow exit channels (dynamics)

**Fig. 8 Schematic of the mechanism and dynamics of the Fe(III)-catalyzed ODA reaction.** Key mechanistic features of the Fe(III)-catalyzed ODA reaction. Summary of our computational mechanism and dynamics results for the Fe(III)-catalyzed ODA reaction.

structures even with considerable C–C bond formation, due to unusual and unrecognized EIE from the change in the metal coordination. Also, dispersion interactions and an oriented external electric field (OEEF) can facilitate the ODA reaction. Therefore, an OEEF may be applied to heterogeneous-type (e.g., MOF[11] and single-atom catalysis[65]) catalysis and artificial metalloenzymes[6,66–70] for the ODA reaction. Moreover, steric and electronic effects were computationally shown to be the key factors responsible for the uncommon C=O chemoselectivity, and this was experimentally verified.

Furthermore, our DFT quasi-classical MD simulations revealed for the first time that the properties of the iron (such as change in its coordination mode and/or spin state) can significantly increase the likelihood of a dynamically stepwise process, broaden its entrance channel, and narrow its exit channel during the reaction dynamics: coordination-, or spin-controlled dynamics. Notably, compared with Brønsted- or Lewis-acid catalysts, the coordination mode and/or spin state of earth-abundant transition metal (e.g., iron) catalysts can be tuned by using different ligands with different electronic and steric effects, which could alter the reaction mechanism as well as reaction dynamics of other reactions. Our proposed metal effects could also be relevant to other metal-catalyzed cycloaddition reactions, and may help the design of homogeneous, heterogeneous[11,65], and artificial biological catalysts[6,69,70] for this synthetically challenging ODA reaction.

## Methods

**DFT calculations**. All calculations were carried out with Gaussian 09 (except DLPNO-CCSD(T)/def2-TZVP(-f) with RIJCOSX and RI-B2PLYP-D3/def2-TZVP methods using ORCA 4.01). The B3LYP-D3/6-31G*, def2-TZVP (Fe) method was used to optimize all the structures in the gas phase. Then, vibration-frequency calculations were carried out at the same level of theory. The effect of the solvent (benzene) was then included in single-point calculations with an implicit solvent model (SMD method, denoted as the SMD B3LYP-D3//B3LYP-D3 method).

Intrinsic reaction coordinate (IRC) calculations on the critical transition states were also performed to obtain their minimum energy path. The geometries of the key structures were also further optimized in solution with the SMD B3LYP-D3 method and the same basis sets. A standard-state concentration (RTln(24.5), or 2.2 kcal mol$^{-1}$) was applied to correct the relative free energy in solution in the text, when the number of molecules is changed. All energies presented are relative Gibbs free energies in solution (at 353.15 K in kcal mol$^{-1}$) above the most stable Fe (III)–porphine complex coordinated to two carbonyl molecules by the SMD B3LYP-D3//B3LYP-D3 method, unless otherwise stated. Furthermore, the secondary deuterium kinetic isotope effect (KIE) and equilibrium isotope effect (EIE) were evaluated using the DFT-computed harmonic frequencies and the Bigeleisen–Mayer equation via PyQuiver code. Validation of B3LYP-D3 results by other DFT methods can be found in Supplementary Tables 1–3. References for those above theoretical methods and software/codes are provided in the Supplementary References.

**Quasi-classical DFT MD simulations**. On-the-fly quasi-classical DFT MD simulations were initiated by normal mode sampling at 353.15 K from the optimized transition states for the reaction of benzaldehyde with 2,3-dimethyl-1,3-butadiene by using Progdyn code[64]. Each trajectory was propagated in both the forward and backward directions with a time step of 1 fs, until either the cycloaddition product formed (the formed C–C and C–O bonds <1.6 Å) or the two reactants separated from each other by more than ~3.3 Å. If the trajectory did not meet either of these two stopping criteria, it could be propagated in one direction only up to 900 fs. The energy and force of all structures at each step were calculated on-the-fly by the B3LYP-D3 (gas phase) and SMD B3LYP-D3 (benzene solution) methods. Only productive trajectories without recrossing events were used for our analysis and discussion.

**Experimental procedure of the KIE study**. Fe(TPP)Cl (0.5 mmol, 352 mg) and AgBF$_4$ (0.5 mmol, 97 mg) were dissolved in dry CH$_2$Cl$_2$ (10 mL) and stirred for 6 h in a dry flask. The reaction mixture was filtered and concentrated to dryness for use. To a flask, the prepared [Fe(TPP)]BF$_4$ (0.050 mmol, 45 mg) was added, followed by the aldehyde (1 mmol, PhCHO:PHCDO = 1:1), the diene (82 mg, 1.0 mmol), and dry benzene (2.0 mL) under the protection of argon. The flask was sealed and stirred under the indicated conditions. The reaction mixture was diluted with hexane (10 mL), passed through a short silica gel pad, and washed with hexane/ethyl acetate = 10/1, concentrated in vacuo. The crude H-1 NMR was determined with CDCl$_3$. The crude product was purified by flash column chromatography using basic Al$_2$O$_3$ (hexane/ethyl acetate, 100:1). KIE is then determined by the ratio of the hydrogen- and deuterium-labeled products.

**Experimental procedure for the formation of E**. Reaction of 2-propenal with 2-phenyl-1,3-pentadiene was performed with the mixing of [Fe(TPP)]BF$_4$ (0.050 mmol, 45 mg), 2-propenal (56 mg, 1 mmol), the diene (144 mg, 1.0 mmol), and dry benzene (4.0 mL) under the protection of argon. The flask was sealed and stirred under the indicated condition. The reaction mixture was diluted with hexane (10 mL), purified by flash column chromatography using basic Al$_2$O$_3$ (hexane/ethyl acetate) to afford product **E2C** in 62% yield.

## Data availability

Data supporting the findings of this study are available within the paper and its Supplementary Information files. All other relevant data are available from the corresponding authors upon reasonable request.

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

## Acknowledgements
We gratefully acknowledge the financial support from the National Natural Science Foundation of China (21672096, 21702095, 21873043 and 21933003), Southern University of Science and Technology, the Shenzhen Nobel Prize Scientists Laboratory Project (C17783101), Guangdong Provincial Key Laboratory of Catalytic Chemistry, and the Natural Science Foundation of Shenzhen Innovation Committee (JCYJ20170817104736009). We thank the Center for Computational Science and Engineering at the Southern University of Science and Technology for partly supporting this work. This paper is dedicated to Professor Kendall N. Houk in recognition of his seminal contributions to computational cycloaddition reactions.

## Author contributions
L.W.C. conceived and designed the project. Y.Y., X.Z., J.L., X.L., and L.W.C. carried out the DFT calculations. L.Z. performed the experimental studies. L.W.C. and C.L. supervised the project. Y.Y., X.Z., X.L., and L.W.C. prepared the paper. All authors analyzed and discussed the results, as well as assisted in paper preparation.

## Competing interests
The authors declare no competing interests.
