## [Peer Review File · Nature Communications]

Reviewers' comments:

Reviewer #1

This is a fine paper, a theoretical treatment of an exciting iron-porphyrin catalyzed hetero-Diels-Alder reaction, discovered by the group of Matsubara and published in J. Am. Chem. Soc. in 2012. Chung et al. now report a very thorough computational analysis and explain the mechanisms of the reactions, involving both doublet, quartet, and sextet spin states. They also measured and computed kinetic isotope effects for these reactions. They also studied both 5- and 6-coordinate pathways and concluded, based on KIE comparisons of theory and experiment, that both operate simultaneously.

This paper is of high quality, but I worry that it will not get the attention it deserves in Nature Communications. It is really primarily of interest to organic, computational and mechanistic chemists. While I would support publication in Nature Communications, I think JACS would be a very (probably more) appropriate vehicle for publication

Reviewer #2 (Remarks to the Author):

This manuscript describes the results of a challenging computational study on a synthetically relevant reaction. Just about every currently "hot" theoretical method is thrown at the reactions in question: eg. dynamics calculations, external electric field calculations. However, in my opinion this leads to a scattered/disjointed presentation. Perhaps the authors can move some information to the SI in pursuit of a focused narrative in the main text?

One additional suggestion: Line 73 - I would certainly mention Carpenter by name here. He was really the first organic chemist to perform such work.

Reviewer #3 (Remarks to the Author):

This manuscript describes a mostly computational study of an iron-catalyzed hetero-Diels-Alder reaction. The combination of regular stationary-point calculations, dynamic trajectories, and an experimental isotope effect measurement make the manuscript a reasonably solid chemical study, if weirdly random (see below) in a number of places. However, all of the most interesting concepts are already in the chemical literature. The current manuscript does not contain the level of advance and novelty that I would expect for Nature Communications, and it is not a close call. I might support a much tighter description of the work in a place like J. Am. Chem. Soc., or with no major modification this could go into J. Org. Chem. appropriately.

Minor issues:

The manuscript sells its science at a level that is too high relative to the science presented, and it otherwise goes over the top on its use of superlative adjectives. There is too much "novel and significant effects" and "unexpectedly large" and "exceptional" and "could help design homogeneous, heterogeneous and biological catalysts for this challenging reaction" and "elegantly" and "seminal," in virtually every paragraph. Now, I believe in selling work and praising the work of others, but the manuscript takes it way too far.

There are more than two prior computational studies of hetero-Diels-Alder reactions.

As a minor pet peeve of this reviewer, "to the best of our knowledge" is one of those phrases that should be left out of the scientific literature. Either one has done enough of a literature search to state something confidently, or not. Yes, one might say something that is wrong, but adding "to the best of our knowledge" would not constitute an excuse anyway.

B3LYP has some problems with dative bonds, and I would not automatically assume that it is the best choice. For a qualitative study it is probably ok, but for definitively stating that some phenomenon is present one would need a bit better backing from either high-level calculations (DLPNO-CCSD(T) is an option these days) or a clear success in predicting an experimental observation. Otherwise, anything special observed is just showing a chemical possibility, not demonstrating that it is correct. It would be especially helpful here to demonstrate that the calculations can accurately predict known energy differences between iron spin states; instead, it seems they can't, and we are left pretty uncertain on what the calculations really show.

Considering the uncertainty in the calculations and the differing molecularity, one really cannot say much about whether the 5- or 6-coordinate pathway is computationally favored.

An interesting question in the trajectory studies is whether the Fe spin state is maintained throughout. Under some circumstances an unrestricted calculation can let a spin state change a lot, and this would mess up the dynamics. Some comment is required.

The OEEF part of the study seems like just throwing in the kitchen sink at the work, with no real focus. Other aspects of the manuscript are like this – the discussion of LUMO's and Mulliken charges and the study of the effect of an imidazole are examples. It gives the manuscript the feel of a random walk through computations rather than having any particular point.

The KIE seems very ordinary compared to other secondary KIEs in Diels-Alder reactions. It is a bit much to take its comparison with predictions that do not fit as a sign of a mixture of mechanisms. I like such comparisons, but one needs to recognize when the evidence is weak. A match between a calculation and an experiment does indeed support the accuracy of the calculation, but calculations are not always accurate, and a match between an experiment and a combination of calculated mechanisms can just as well be ascribed to an exercise in numerology.

For the dynamics, I personally do not find the subject to be interesting unless it somehow affects experimental observations. One could imagine this if the isotope effect had supported full C-C bond formation reversibly before C-O bond formation, but it doesn't. Issues like "dynamically concerted" versus "dynamically stepwise" based on arbitrarily chosen time gaps between bond formations seem obscure without experimental ramifications.

Reviewer #4 (Remarks to the Author):

Comments:

Hetero-Diels-Alder (HDA) reaction is an important synthetic method for many natural products but it suffers from the challenge towards unactivated substrates. In 2012, a cationic Fe(III) porphyrin catalyst was reported to address this challenge with great selectivity towards unactivated aldehydes and dienes (JACS 2012, 134, 5512). In this paper, the authors conducted DFT calculations and quasi-classical MD simulations along with two mechanistic experiments to investigate the mechanism and dynamics of this Fe-catalyzed oxa-Diels-Alder reaction (ODA). Results gained from these studies can advantage the mechanistic understanding and inform design of catalysts for this class of reactions. According to the results, both 5-coordinate and 6-coordinate mode pathway in quartet and sextet spin states of Fe (two-state two-mode reactivity) are feasible for this ODA reaction through asynchronous mechanism. The authors attribute the unusually large secondary KIE values and unusual EIE value to the change in metal coordination of 5-coordinate pathway during reaction, which was supported by corresponding mechanistic experiments. Chemoselectivity on C=O bond of unactivated aldehydes by Fe catalyst was also verified by DFT calculation and related experiment. The results from quasi-classical MD simulations suggest that change in coordination mode and/or spin state of Fe catalyst as well as the addition of an OEEF

can tune the probability of dynamically-stepwise mechanism of this ODA reaction despite its concerted asynchronous transition state. Overall, the performed calculations and experiments are thorough, and the conclusions made in the manuscript are well supported by the results. Thus, I am supportive of potential publication in this journal after the following issues have been addressed:

Major issues:

1. The counterion has found experimentally to have a pronounced effect on the reactivity of the system. As shown in Table 1 in the experimental paper (JACS 2012, 134, 5512), using the same catalyst core and substrate and solvent, the counterions Cl⁻, OTf⁻, and PF6⁻ failed to give any yield (<1 %) while BF4 gave 92% yield. I understand the modeling the counterion is challenging and likely beyond the scope of this paper. However, I was a bit disappointed that the authors omitted the discussion on this important component in reactivity in the manuscript. The authors should clearly discuss the counterion effect in the reaction and provide some sort of calculations addressing the effect of the counterion on the reactivity. Moreover, looking closely at Figure 3, the 5-coordinate pathway seems to be reversible. The authors should discuss the implications of this reversibility. I was surprised that the authors failed to mention anything about the energies of the products.

1. Fig. 2 contains too much information that is difficult to follow. Authors need to clean this figure and put the less necessary details in Supporting information (e.g. interatomic distances that are not important or mentioned in the discussion in manuscript, and data obtained from other methods). For example, does the molecular geometry shown belong to quartet or sextet Fe species? Differentiation needs to be made and the geometry of the other Fe species with different spin state can be put in supporting information.

2. Page 8, line 168, "dispersion attraction was found to lower these reaction barriers" is confusing. What exactly do the authors mean by such statement and what is the origin of this "barrier lowering?". Does the including of dispersion (D3) change the structure of the reactant? Of the TS? in such way that, in comparison the gas phase structures and geometries, the barrier is lowered? If authors have other information supporting this statement, please provide.

3. Exploring and discussing the "Metal Effect" distracts from the paper. It is more important to focus on the counterion effect since it has a greater impact in reactivity for the same system. In this case, Ru could also be sensitive to solvent, counterion, etc. thus the comparison with Fe seems less impactful and more prone to drawing erroneous computational conclusions.

4. Exploring the "Biological Effect" section could be moved to the SI or mentioned in the conclusions as potential impact. As it stands, it is very distracting from an already very complex computational and experimental paper.

5. Fig. 6 discusses chemoselectivity with Fe complex but in the text the uncatalyzed reaction is also discussed. The authors should also include the uncatalyzed values in this figure and make it more easy to follow.

6. Fig. 8 data from quasi-classical MD simulations is difficult to follow and overall distracts from the paper. Figure 7 is sufficient to highlight the key aspects from these calculations. Authors need to put the less important details (Figure 8) in SI and present the results in a cleaner way in the manuscript.

7. Fig. 9 contains too much information as a summary figure of the manuscript. It needs to be

cleaned. For example, "porphyrin ligand as an auxiliary electron reservoir" can be deleted from the figure.

8. Figure 7. Please add labels DIRECTLY to the Figure! The authors should not expect readers to read a whole paragraph as the caption to understand the figure!

9. Line 166, the authors mentioned "representing two-state mode reactivity". However, the data supporting this statement is very weak at best since, as shown in Figure 3, the LOWEST energy pathway starts from QUARTET and PROCEEDS VIA QUARTET. There is no need to make mentioned of such two-state reactivity especially when the data supporting such statement is not strong.

Minor issues:

1. Line 25, "a α,β -unsaturated aldehyde" should be "an α,β -unsaturated aldehyde".

2. Page 4, line 88-89, the solvent used in with SMD model and the corresponding atoms applied with basis set should be mentioned in manuscript instead of only in Supporting Information.

5. Page 5, line 120, "d5 Fe(III) metal" should be "d5 Fe(III) metal".

6. In Fig. 2, authors need to specify that "LUMO" refers to $\pi^*(C=O)$ orbital but not the LUMO of Fe catalytic system (which might partly consist of orbitals of Fe). Further, the authors mentioned the significant LUMO lowering but failed to compare with other LUMO activation models.

7. In Fig.3, do authors have information about the relative energy of product dissociating from Fe center after reaction and the rebound of another aldehyde molecule?

9. In Figure S2 (in SI), illustration of exo ODA reaction is missing (neither molecular structure not images). Moreover, labels and energy values are overlapping with one another in Figure S2, making it difficult to follow. Please modify this figure.

10. In Fig. 5(a), the authors used comma "," in label "2,4IA" to indicate different spin states on the left part, while on the right part, "/" was used in label "2TS1Aendo/4TS1Aendo" to differentiate different spin states. In Fig. 5(b) and Fig. 2, comma "," were usually used. Please modify the labels to stay consistent.

11. Page 13, line 279, "effect of using one axial ..." should be "effects of using one axial ..."

12. Page 19, line 409-410, what do "some trajectories" indicate? More details need to be provided for the statement here to indicate what trajectories show such "intermediate" region and which part of Figure S24 should we look at for these information. Further, in the text, it is mentioned that a total of 1310 trajectories were ran. The authors need to provide the breakdown of these trajectories. How many for solvent? How many for gas phase? How many for uncatat and cat?

13. Page 19, line 423, is "~21-87%" the probability in the absence of an OEEF? Statement needs

to be clearer here to make better comparison between probability in the absence and presence of OEEF.

14. In Fig.9, does "C-C: 1.86-1.93" in 6-coordination mode indicate the distance between two C atoms? Unit needs to be added and clarification should be made on which two carbon atoms are indicated. Also, colors in Fig. 9 are too varied that it looks not clean enough.

15. Line 177 "were found to be less stable". This should be clarified. The authors can simply state that it is found higher in energy NOT less stable since "stability" is meaningless. There is kinetic or thermodynamic stability. Which are the authors referring to?

16. Page 20, line 436-437, the statement "synthetically challenging Fe-catalyzed ODA reaction of the unactivated substrates" looks confusing because ODA reaction is synthetically challenging towards unactivated substrates while in this study Fe catalyst can address this issue. Please modify this statement.

17. Figure 4. The authors should include error bars for the experimental KIE.

Response to Reviewers' comments

We sincerely appreciate the four reviewers for their very useful/constructive comments and suggestions on our paper. This document summarizes our responses to all their questions/suggestions and points to the changes we have made in the revised manuscript and SI. Such changes include additional technical information, such as the suitability of B3LYP to describe energies for different iron's spin states; high-level calculations by DLPNO-CCSD(T) method for the key intermediates and transition states; calculations on the key KIE and EIE values by six other DFT methods; additional calculations and discussion on the effects of the charge on the Fe catalysts; additional calculations about energies of different spin states for the related Fe(III) complex with coordination of two acetone molecules. We have also revised some figures and moved some information (e.g. "The Metal Effect" and "The Biological Ligand Effect") to the SI to make the manuscript clear and coherent.

The comments from the reviewers (in an italic form)

Our response (in a plain form)

Our revised part in the text and SI (italic and yellow highlight)

Reviewer #1 (Remarks to the Author):

Comment: *This is a fine paper, a theoretical treatment of an exciting iron-porphyrin catalyzed hetero-Diels-Alder reaction, discovered by the group of Matsubara and published in J. Am. Chem. Soc. in 2012. Chung et al. now report a very thorough computational analysis and explain the mechanisms of the reactions, involving both doublet, quartet, and sextet spin states. They also measured and computed kinetic isotope effects for these reactions. They also studied both 5- and 6-coordinate pathways and concluded, based on KIE comparisons of theory and experiment, that both operate simultaneously.*

This paper is of high quality, but I worry that it will not get the attention it deserves in Nature Communications. It is really primarily of interest to organic, computational and mechanistic chemists. While I would support publication in Nature Communications, I think JACS would be a very (probably more) appropriate vehicle for publication.

Response: We would like to sincerely thank the reviewer's positive comments. Both *Nature Communications* and *JACS* are high-quality and comprehensive journals. Although our study focuses on the mechanism of homogeneous Fe-catalyzed organic (ODA) reaction, our computational results and

discussion also mentioned and implied this ODA reaction for heterogeneous catalysts (MOF, single-atom catalysis) and artificial metalloenzymes.

Reviewer #2 (Remarks to the Author):

Comment: *This manuscript describes the results of a challenging computational study on a synthetically relevant reaction. Just about every currently "hot" theoretical method is thrown at the reactions in question: eg. dynamics calculations, external electric field calculations. However, in my opinion this leads to a scattered/disjointed presentation. Perhaps the authors can move some information to the SI in pursuit of a focused narrative in the main text?*

Response: We are sincerely grateful for the reviewer's positive comments and helpful suggestions. Based on your suggestions, we have moved "The Metal Effect" and "The Biological Ligand Effect" sections, the *external electric field* results in **Fig. 8** to the revised SI.

Comment: *One additional suggestion: Line 73 - I would certainly mention Carpenter by name here. He was really the first organic chemist to perform such work.*

Response: Thank you so much for your excellent suggestion and reminder. We fully agree with you. In addition, we would also like to mention Prof. Hase, who contributed several important and early dynamics works on SN2 reactions. We have revised the sentence and added the new ref. 60 in the revised manuscript as follows:

"In addition, Carpenter, Hase, Houk, Singleton, Tantillo and other groups carried out quasi-classical molecular dynamics"

"60. S. L. Debbert, B. K. Carpenter, D. A. Hrovat, W. T. Borden, The Iconoclastic Dynamics of the 1,2,6-Heptatriene Rearrangement. J. Am. Chem. Soc. 2002, 124, 7896."

Reviewer #3 (Remarks to the Author):

This manuscript describes a mostly computational study of an iron-catalyzed hetero-Diels-Alder reaction. The combination of regular stationary-point

calculations, dynamic trajectories, and an experimental isotope effect measurement make the manuscript a reasonably solid chemical study, if weirdly random (see below) in a number of places. However, all of the most interesting concepts are already in the chemical literature. The current manuscript does not contain the level of advance and novelty that I would expect for Nature Communications, and it is not a close call. I might support a much tighter description of the work in a place like J. Am. Chem. Soc., or with no major modification this could go into J. Org. Chem. appropriately.

Response: We sincerely appreciate the reviewer's review and comments. Although many interesting concepts were already documented in the chemical literature, **a few new or unusual chemical findings were reported in our study (e.g. unusual large secondary KIE for the 5-coordinate pathway caused by unrecognized EIE from the change in the metal coordination; dynamics controlled by coordination and/or spin state of the iron metal as well as by an external electric field; the porphyrin ligand as an auxiliary electron reservoir in the reaction)**. Both *Nature Communications* and *JACS* are high-quality and comprehensive journals. Based on all of your questions and suggestions, we have performed additional calculations as well as made significant revisions. We hope that you are satisfied with our revisions.

Minor issues:

Comment: *The manuscript sells its science at a level that is too high relative to the science presented, and it otherwise goes over the top on its use of superlative adjectives. There is too much "novel and significant effects" and "unexpectedly large" and "exceptional" and "could help design homogeneous, heterogeneous and biological catalysts for this challenging reaction" and "elegantly" and "seminal," in virtually every paragraph. Now, I believe in selling work and praising the work of others, but the manuscript takes it way too far.*

Response: We appreciate the reviewer's kind comment and suggestion. We have deleted or changed most of these words (adjectives). Please see the highlighted changes in the revised manuscript for the details.

Comment: *There are more than two prior computational studies of hetero-Diels-Alder reactions.*

Response: We fully agree with the reviewer that there are some other previous computational studies of (almost metal-free) hetero-Diels-Alder reactions, compared to much more computational studies on Diels-Alder

reactions. As far as we found, there is only one computational study on the challenging 1st row transition metal (Cobalt) catalyzed hetero-Diels-Alder reaction (the *Angew. Chem. Int. Ed.*, **2005**, 2524. paper we cited). It should be noted that our work is the first computational report on complicated Fe-hetero-Diels-Alder reaction as well as should be the first DFT quasi-classical MD for hetero-Diels-Alder reaction. Due to limited numbers of references in *Nature Communications*, we cited the two papers. If you want to cite 1-2 more references, we are grateful to add 1-2 more references.

Comment: *As a minor pet peeve of this reviewer, “to the best of our knowledge” is one of those phrases that should be left out of the scientific literature. Either one has done enough of a literature search to state something confidently, or not. Yes, one might say something that is wrong, but adding “to the best of our knowledge” would not constitute an excuse anyway.*

Response: We thank a lot for your helpful comment. Based on your suggestion, we have deleted “to the best of our knowledge” in the revised manuscript.

Comment: *B3LYP has some problems with dative bonds, and I would not automatically assume that it is the best choice. For a qualitative study it is probably ok, but for definitively stating that some phenomenon is present one would need a bit better backing from either high-level calculations (DLPNO-CCSD(T) is an option these days) or a clear success in predicting an experimental observation. Otherwise, anything special observed is just showing a chemical possibility, not demonstrating that it is correct. It would be especially helpful here to demonstrate that the calculations can accurately predict known energy differences between iron spin states; instead, it seems they can't, and we are left pretty uncertain on what the calculations really show.*

Response: We sincerely thank your comments and questions. We agree that quantitative description about energy gap between different spin states for Fe complexes has long been one of the holy grails in computational chemistry. B3LYP has been widely used because of many successful applications to heme- and non-heme systems in many research groups (e.g. Himo, Morokuma, Neese, Shaik, Siegbahn, Thiel and de Visser). Moreover, apart from B3LYP method, we used different and common DFT methods for Fe complexes. Different DFT methods qualitatively gave the similar mechanistic results. Notably, we already mentioned in our manuscript that “several Fe(III) porphyrin complexes with two axial oxygen ligands were experimentally found to have admixed spin states (quartet and sextet states)” (e.g. *Nat. Commun.*

2018, 4750 and references cited therein). Hence, the quartet and sextet states for the related Fe(III) porphyrin complexes were experimentally found to be quite similar in energy indeed. Based on your suggestion, we have performed additional energy calculations of the key intermediates and transition states to form **A** by the DLPNO-CCSD(T) and B2PLYP-D3 methods (see **Supplementary Table 2**). Additionally, PBE0-D3, B3PW91-D3, PW6B95D3, OLYP-D3, TPSSh-D3, OPBE-D3 and ω B97XD methods have also been used to optimize the key intermediates and transition states in solution, and their results can support the B3LYP-D3 results (see **Supplementary Table 2**). In addition, a Fe(III) porphyrin complex coordinating with two acetone as axial ligands which was recently experimentally found to have admixed spin states (*Nat. Commun.* **2018**, 4750) was used to test different DFT and DLPNO-CCSD(T) methods in our additional calculations (see **Supplementary Table 17**). However, compared to the experimental observation with the admixture spin states, the DLPNO-CCSD(T) method was found to significantly over-stabilize the high spin state ($\Delta G_{S-Q} = \sim -11.8$ kcal/mol, see **Supplementary Table 17**). Recently, Harvey group performed their multireference approach CASPT2/CC to calculate the quintet-triplet gaps of a series of non-heme Fe(IV)=O species (*J. Chem. Theory Comput.* **2019**, 4297; *J. Chem. Theory Comput.* **2019**, 922). Their work also concluded that “current implementations of the local coupled-cluster method are not sufficiently accurate. DLPNO-CCSD(T) systematically over-stabilizes the quintet state”. Unfortunately, the practical, advanced and state-of-the-art DLPNO-CCSD(T) method predicted the wrong ground state. Furthermore, Radoń performed CCSD(T) calculations on simplified heme-type Fe(III) complexes and compared with different DFT methods (*J. Chem. Theory Comput.* **2014**, 2306). This study summarized that “Although the DFT results are highly functional-dependent, it is shown that the spin-state energetics of a full heme model and its simplified mimic remain in a good linear correlation.” As shown the below Figures taken from this paper, PBE0 was shown to be the best method for energy gap between the quartet and sextet states for the heme-like Fe(III) cases. It is also in agreement with our new calculations on the Fe(III)

porphyrin complex coordinating with two acetone that PBE0 qualitatively reproduce the observed admixture states (**Supplementary Table 17**). Our PBE0 results did support our B3LYP results that the key transition states have the similar energy for the quartet and sextet states.

In the revised manuscript, we have revised the results and discussion as well as new refs. 61-62 as follows:

“Importantly, these results, which are also qualitatively supported by different DFT functionals (the B3PW91-D3, PBE0-D3, ω B97XD, PW6B95-D3, OLYP-D3, and OPBE-D3 methods, see Supplementary Table 2)”

“The state-of-the-art DLPNO-CCSD(T) method was also employed to evaluate the relative energy of the critical structures and the related Fe(III) complex coordinating with two acetone as axial ligands⁴⁸ (see Supplementary Tables 2 and 17 as well as Supplementary Figure 10). However, compared to the observed admixture states,⁴⁸ this method was found to significantly overstabilize the high spin states⁶¹ than the quartet state ($\Delta G_{S-Q} = \sim -11.8$ kcal/mol). Whereas, the PBE0 method was a good method to describe this energy gap⁶² and also supported the SMD B3LYP-D3 results in this system.”

“61. M. Feldt, Q. M. Phung, K. Pierloot, R. A. Mata, J. N. Harvey. Limits of Coupled-Cluster Calculations for Non-Heme Iron Complexes. J. Chem. Theory Comput. 2019, 15, 922.”

“62. M. Radoń, Spin-State Energetics of Heme-Related Models from DFT and Coupled Cluster Calculations. J. Chem. Theory Comput. 2014, 10, 2306.”

Also, we have revised the below discussion and tables in the revised SI:

“We have performed additional energy calculations of the key intermediates and transition states to form A by the DLPNO-CCSD(T), B2PLYP-D3 PBE0-D3, B3PW91-D3, PW6B95D3, OLYP-D3, TPSSh-D3, OPBE-D3 and

ω B97XD methods. In addition, a Fe(III) porphyrin complex coordinating with

two acetone as axial ligands which was recently experimentally found to have admixed spin states (Nat. Commun. **2018**, 4750) was used to test different DFT and DLPNO-CCSD(T) methods in our additional calculations.”

“However, compared to the experimental observation (admixture quartet and sextet states), the DLPNO-CCSD(T) method was found to significantly over-stabilize the high spin state ($\Delta G_{S-Q} = \sim -11.8$ kcal/mol, see Supplementary Table 17). Recently, Harvey and coworkers performed their multireference approach CASPT2/CC to calculate the quintet-triplet gaps of a series of non-heme Fe(IV)=O species (J. Chem. Theory Comput. **2019**, 4297; J. Chem. Theory Comput. **2019**, 922). Their works also concluded that “current implementations of the local coupled-cluster method are not sufficiently accurate. DLPNO-CCSD(T) systematically over-stabilizes the quintet state”. Unfortunately, the practical, advanced and state-of-the-art DLPNO-CCSD(T) method predicted the wrong ground state. Furthermore, Radoń performed CCSD(T) calculations on simplified heme-type Fe(III) complexes and compared with different DFT methods (J. Chem. Theory Comput. **2014**, 2306). This study summarized that “Although the DFT results are highly functional-dependent, it is shown that the spin-state energetics of a full heme model and its simplified mimic remain in a good linear correlation.” PBE0 method was shown to be the best method for energy gap between the quartet and sextet states for the heme-like Fe(III) cases in this study. It is also in agreement with our new calculations on the Fe(III) porphyrin complex coordinating with two acetone that the PBE0-D3 method qualitatively reproduce the observed admixture states (Supplementary Table 17). Our PBE0-D3 results did support our B3LYP-D3 results that the key transition states have the similar energy for the quartet and sextet states (see Supplementary Table 2).”

Supplementary Table 2. The relative energies (in kcal/mol at 353.15 K) for the key intermediates and transition states of **A** in quartet (Q) and sextet (S) states optimized by other SMD DFT methods and single-point calculations by the DLPNO-CCSD(T) and B2PLYP-D3 methods in gas phase based on the SMD B3LYP-D3-optimized structures. **Notably, the standard state correction was applied here.**

	LCCSD(T) (ΔE) ^a ΔG ^b	B2PLYP-D3 (ΔE) ^a ΔG ^b	PBE0-D3	B3PW91-D3	PW6B95-D3	BP86-D3
⁴ IA	(0.0)0.0	(0.0)0.0	0.0	0.0	0.0	0.0
⁶ IA	(-9.8)-11.2	(4.6)3.2	2.9	7.7	5.2	1.7
⁴ TS1A _{endo}	(5.5)24.4	(6.1)24.9	20.7	22.1 ^c	24.7	14.4 ^d
⁶ TS1A _{endo}	(-8.2)7.7	(7.5)23.4	19.2	21.8	25.4	26.3
⁴ TS2A _{endo}	(25.2)30.4	(23.3)28.4	20.8	21.1	24.6	13.7
⁶ TS2A _{endo}	(13.3)15.1	(25.3)27.1	18.4	20.3	28.2	26.7

	M06-L	B3LYP*-D3	OLYP-D3	TPSSh-D3	OPBE-D3	ωB97XD
⁴ IA	0.0	0.0	0.0	0.0	0.0	0.0
⁶ IA	-4.5	15.4	6.0	8.9	4.4	6.4
⁴ TS1A _{endo}	26.3	18.2	18.7	20.9	14.6	21.6
⁶ TS1A _{endo}	19.4	28.7	19.8	27.2	17.1	24.3
⁴ TS2A _{endo}	25.0 ^e	19.9	18.4	20.1	14.3	23.2
⁶ TS2A _{endo}	18.1	31.1	21.0	28.1	17.0	21.9

a. The relative electronic energies (in kcal/mol) in gas phase based on the SMD B3LYP-D3-optimized structures. b. The relative free energies (in kcal/mol) in solution based on the SMD B3LYP-D3 method. c. One additional small imaginary frequency (-6.96) exists. d. One additional small imaginary frequency (-9.27) exists. e. One additional small imaginary frequency (-10.92) exists.

Supplementary Table 17. The relative energies (in kcal/mol at 298.15 K) of the key structures to form **G** with replacing PhCHO of acetone in acetone solution by the SMD B3LYP-D3 method and other methods based on the SMD B3LYP-D3 optimized structures.

	B3LYP-D3	LCCSD(T) ^a (ΔE)	LCCSD(T) ^b ΔG	PBE0-D3	B3PW91-D3
² IG	13.9	-	-	16.1	14.4
⁴ IG	0.0	(0.0)	0.0	0.0	0.0
⁶ IG	6.4	(-10.4)	-11.8	1.3	4.9
² IG _{4Ph}	13.4	-	-	17.1	14.7
⁴ IG _{4Ph}	0.0	-	-	0.0	0.0
⁶ IG _{4Ph}	5.8	-	-	0.8	4.5
	PW6B95-D3	M06-L	OLYP-D3	OPBE-D3	ωB97XD
² IG	14.8	12.9	16.8	16.0	11.6
⁴ IG	0.0	0.0	0.0	0.0	0.0
⁶ IG	4.4	-3.8	6.6	5.5	5.4
² IG _{4Ph}	16.3	16.1	13.9	14.9	14.8
⁴ IG _{4Ph}	0.0	0.0	0.0	0.0	0.0
⁶ IG _{4Ph}	4.1	-4.4	6.3	5.3	5.4

a. The relative electronic energies (in kcal/mol) in gas phase based on the SMD B3LYP-D3-optimized structures. b. The relative free energies (in kcal/mol) in solution based on the SMD B3LYP-D3 method.

@Xx: Radoń, M. Spin-State Energetics of Heme-Related Models from DFT and Coupled Cluster Calculations. *J. Chem. Theory Comput.* **2014**, *10*, 2306.

Xx: Phung, Q. M. Feldt, M. Harvey, J. N. Pierloot, K. Toward Highly Accurate Spin State Energetics in First-Row Transition Metal Complexes: A Combined CASPT2/CC Approach. *J. Chem. Theory Comput.* **2018**, *14*, 2446.

xx. Phung, Q. M. Martín-Fernández, C. Harvey, J. N. Feldt, M. Ab Initio Calculations for Spin-Gaps of Non-Heme Iron Complexes. *J. Chem. Theory Comput.* **2019**, *15*, 4297.

xx. Feldt, M. Phung, Q. M. Pierloot, K. Mata, R. A. Harvey, J. N. Limits of Coupled-Cluster Calculations for Non-Heme Iron Complexes. *J. Chem. Theory Comput.* **2019**, *15*, 922.

XX. Kepp, K. P. Theoretical Study of Spin Crossover in 30 Iron Complexes. *Inorg. Chem.* **2016**, *55*, 2717.

Xx: Cirera, J. Via-Nadal, M. Ruiz, E. Benchmarking Density Functional Methods for Calculation of State Energies of First Row Spin-Crossover Molecules. *Inorg. Chem.* **2018**, *57*, 14097.

Comment: *Considering the uncertainty in the calculations and the differing molecularity, one really cannot say much about whether the 5- or 6-coordinate pathway is computationally favored.*

Response: Thanks a lot for your comment. Different and common DFT methods qualitatively gave the same mechanistic results in our study. In addition, the computed KIE value for the 6-coordinate pathway by the various DFT methods is always lower than the measured KIE value. A better KIE agreement between the calculations and experiment were found when combining with the computed large KIE value for the 5- and 8-coordinate pathways, suggesting both working pathways.

Comment: *An interesting question in the trajectory studies is whether the Fe spin state is maintained throughout. Under some circumstances an unrestricted calculation can let a spin state change a lot, and this would mess up the dynamics. Some comment is required.*

Response: This is a very interesting question. Before we performed the calculations, we also wondered possibility of changing spin density on the metal during the reaction, as single electron transfer was reported in some computational studies for other reactions. We have added a new figure (**Supplementary Figure 27**) to show the distribution of Fe spin in all geometries generated by those productive trajectories (from reactant to product). The Fe spin density does not change significantly throughout the dynamics. We have revised the below discussion in the revised manuscript:

“Throughout those Fe-catalyzed trajectories, the spin density of Fe was found to have only small changes (Supplementary Figure 27).”

We have also added the below new figure in the revised SI:

Supplementary Figure 27. Distribution of Fe spin density in the 6-coordinate Quartet Fe complexes (left), 5-coordinate Quartet (middle) and Sextet (left) Fe complexes accessed by all the productive trajectories for the (A) gas phase simulations, solution phase (SMD) simulations (B) in absence of oriented external electric field and (C) in presence of oriented external electric field.

Comment: The OEEF part of the study seems like just throwing in the kitchen sink at the work, with no real focus. Other aspects of the manuscript are like this – the discussion of LUMO's and Mulliken charges and the study of the effect of an imidazole are examples. It gives the manuscript the feel of a random walk through computations rather than having any particular point.

Response: We sincerely thank the reviewer for this helpful comment. We have moved some of the OEEF part, “The Metal Effect”, “The Biological Ligand Effect” and the discussion of LUMO's and Mulliken charges to the SI.

Comment: The KIE seems very ordinary compared to other secondary KIEs in Diels-Alder reactions. It is a bit much to take its comparison with predictions that do not fit as a sign of a mixture of mechanisms. I like such comparisons,

but one need to recognize when the evidence is weak. A match between a calculation and an experiment does indeed support the accuracy of the calculation, but calculations are not always accurate, and a match between an experiment and a combination of calculated mechanisms can just as well be ascribed to an exercise in numerology.

Response: We appreciate the reviewer for your comment. Calculations usually gave reliable secondary KIE value in many previous combined computational and experimental studies on the metal-free DA and HDA reactions (we cited some of them), as quantum tunneling is not important in that case. Interestingly, our computed secondary KIE value for the 6-coordinate pathway by different DFT methods is always less than the measured value, whereas that value for the 5-coordinate pathway is larger than the measured value. Only one pathway gave a large discrepancy and did not give good agreement with the experiment. However, consideration of KIE contributed by both pathways leads to better agreement with the measured value, when assuming similar contribution to both pathways and spin states (quartet and sextet). The total KIE is the sum of the KIE value for each pathway/state time its probability ($KIE_{total} = \sum P_i KIE_i$; probability is determined by the relative free energy). We should stress that our key finding is the unusual equilibrium isotope effect (EIE) from the change in the metal coordination, which renders both pathways to have quite different KIE values even for very similar transition state structures. We have also performed additional KIE calculations by using a few more methods (see **Supplementary Table 14**). Such large KIE and EIE effects which were supported by different DFT method are unusual and usually unrecognized. All the DFT results indicated the five-coordinate pathway has larger KIE than the experimental value and the computed EIE value is not small (1.069-1.103), while the six-coordinate pathway has smaller KIE than the experimental value. These results further supported that both the six- and five-coordinate mode pathways should be involved in the Fe-catalyzed ODA reaction. We have added **Supplementary Table 14** in the revised SI.

Supplementary Table 14. Computed secondary deuterium (k_H/k_D) KIE and EIE results for the formation of **A** in quartet (Q) and sextet (S) states optimized by other DFT methods and SMD method in solution at 353.15 K.

	B3LYP-D3	PBE0-D3	B3PW91-D3
EIE			
⁴ IA → ⁴ IVA	1.069	1.082	1.080
KIE			
⁴ TS1A _{endo}	0.905	0.902	0.918 ^b
⁶ TS1A _{endo}	0.902	0.887	0.914

⁴ TS2A _{endo}	0.985	0.975	0.992
⁶ TS2A _{endo}	N/A ^a	0.950	0.972
	M06-L	B3LYP*-D3	OLYP-D3
EIE			
⁴ IA → ⁴ IVA	1.103	1.068	1.079
KIE			
⁴ TS1A _{endo}	0.900	0.877	0.888
⁶ TS1A _{endo}	0.880	0.877	0.887
⁴ TS2A _{endo}	0.951 ^c	0.975	0.951
⁶ TS2A _{endo}	0.934	0.962	N/A ^a

a. A significant numerical problem was found. b. One additional small imaginary frequency (-6.96) exists. c. One additional small imaginary frequency (-10.92) exists.

Comment: For the dynamics, I personally do not find the subject to be interesting unless it somehow affects experimental observations. One could imagine this if the isotope effect had supported full C-C bond formation reversibly before C-O bond formation, but it doesn't. Issues like "dynamically concerted" versus "dynamically stepwise" based on arbitrarily chosen time gaps between bond formations seem obscure without experimental ramifications

Response: Thank you so much for your comments. As mentioned in our manuscript, recent DFT quasi-classical molecular dynamics studies have offered us several new mechanistic concepts for some organic and enzymatic reactions and attracted considerable attention from chemistry community. Also, our study revealed for the first time that different coordination mode and/or spin state of the Fe metal can obviously affect reaction dynamics from concerted asynchronous transition states. Notably, molecular dynamics studies reveal the time-resolved details of bond formation and breaking on the reaction dynamics, which are very important in chemistry. However, it is very challenging to reveal dynamics of many rare and ultrafast chemical events in experiments. Reaction dynamics for some ultrafast photochemical reactions or some few-atom thermal reactions using experimental means were mostly reported only. We agree with you that the measured and computed KIE should relate to the transition state (TS) geometry, which mainly determine the reaction rate and, thus, KIE. Our current and other recent dynamics studies show interesting post-TS dynamics (reaction dynamics after passing transition state region), which should not significantly affect the reaction rate and KIE.

The proposed time gap criterion (~60 fs at 300 K) is not arbitrary, and is

quite reasonable based on timing of one complete C-C bond vibrations proposed in the recent Houk paper (*PNAS*, **2012**, 10358), which is also consistent with the life time ($h/k_B T$) of transition state defined by the Eyring kinetic theory. This criterion is in fact helpful in dynamical simulations, since it serves as a parameter to distinguish “dynamical concerted” and “dynamical stepwise” trajectories and measure the asynchronicity of a concerted reaction. In addition, this time criterion has been widely used to describe the dynamics of many pericyclic reactions (e.g., Diels-Alder reactions, $SpnF$ catalyzed-enzymatic reactions) and dimethyldioxirane C-H oxidation reaction. Notably, in this work, a significant portion of the 5-coordinate Fe-catalyzed ODA trajectories, especially in the presence of OEEF, resides in an intermediate region for hundreds of femtoseconds before going to the product. It might be possible to detect or trap the related intermediate species in the presence of OEEF in the future.

Reviewer #4 (Remarks to the Author):

Comments:

Hetero-Diels-Alder (HDA) reaction is an important synthetic method for many natural products but it suffers from the challenge towards unactivated substrates. In 2012, a cationic Fe(III) porphyrin catalyst was reported to address this challenge with great selectivity towards unactivated aldehydes and dienes (JACS 2012, 134, 5512). In this paper, the authors conducted DFT calculations and quasi-classical MD simulations along with two mechanistic experiments to investigate the mechanism and dynamics of this Fe-catalyzed oxa-Diels-Alder reaction (ODA). Results gained from these studies can advantage the mechanistic understanding and inform design of catalysts for this class of reactions. According to the results, both 5-coordinate and 6-coordinate mode pathway in quartet and sextet spin states of Fe (two-state two-mode reactivity) are feasible for this ODA reaction through asynchronous mechanism. The authors attribute the unusually large secondary KIE values and unusual EIE value to the change in metal coordination of 5-coordinate pathway during reaction, which was supported by corresponding mechanistic experiments. Chemoselectivity on C=O bond of unactivated aldehydes by Fe catalyst was also verified by DFT calculation and related experiment. The results from quasi-classical MD simulations suggest that change in coordination mode and/or spin state of Fe catalyst as well as the addition of an OEEF can tune the probability of dynamically-stepwise mechanism of this ODA reaction despite its concerted asynchronous transition state. Overall, the performed calculations and experiments are thorough, and the conclusions made in the manuscript are well supported by the results. Thus, I am

supportive of potential publication in this journal after the following issues have been addressed:

Response: We appreciate your positive and constructive comments very much.

Major issues:

Comment: 1. The counterion has found experimentally to have a pronounced effect on the reactivity of the system. As shown in Table 1 in the experimental paper (JACS 2012, 134, 5512), using the same catalyst core and substrate and solvent, the counterions Cl⁻, OTf⁻, and PF₆⁻ failed to give any yield (<1 %) while BF₄⁻ gave 92% yield. I understand the modeling the counterion is challenging and likely beyond the scope of this paper. However, I was a bit disappointed that the authors omitted the discussion on this important component in reactivity in the manuscript. The authors should clearly discuss the counterion effect in the reaction and provide some sort of calculations addressing the effect of the counterion on the reactivity. Moreover, looking closely at Figure 3, the 5-coordinate pathway seems to be reversible. The authors should discuss the implications of this reversibility. I was surprised that the authors failed to mention anything about the energies of the products.

Response: Thank you so much for your important comments and questions. Based on your concerns, we have performed additional calculations to examine and compare the reactivity of three Fe(III) catalysts. Cl⁻ or OTf⁻ was known to have (very) weak ability of being a counterion and, as shown in our new calculations (the SMD B3LYP-D3 method), it forms a direct chemical bond with the metal as an axial ligand to afford the neutral Fe(III) complexes. Whereas, non-coordinating anions BF₄⁻ and SbF₆⁻ were known to be very good counterions and, thus, the charge of the Fe(III) catalyst can be regarded to be cationic (i.e. higher Lewis acidic metal). Overall, our additional calculations revealed that the neutral Fe complexes have a much high barrier (29.9-36.0 kcal/mol) for the ODA reaction, which qualitatively explains the pronounced effect on different reactivity for different Fe complexes. We have added these below results and discussion in a new section (“The Charge Effect of the Fe complexes”) of the revised manuscript as well as added **Supplementary Figure 11** and **Supplementary Table 49** in the revised SI:

“The Charge Effect of the Fe complexes. Owing to lower Lewis acidity of the Fe metal, our additional SMD B3LYP-D3 results showed that the two neutral Fe(III) complexes with coordination of one anionic Cl⁻ or OTf⁻ ligand in the five-coordinate mode ^{4,6}IVA(Cl⁻) and ^{4,6}IVA(OTf⁻) are the most stable species

(without coordination of a PhCHO substrate) before ODA reaction (see Supplementary Figure 11). ${}^6\text{IVA}(\text{Cl}^-)$ and ${}^4\text{IVA}(\text{OTf}^-)$ are just about 1.3 and 1.4 kcal/mol lower in free energy than ${}^4\text{IVA}(\text{Cl}^-)$ and ${}^6\text{IVA}(\text{OTf}^-)$, respectively. Coordination of one PhCHO to $\text{IVA}(\text{Cl}^-)$ or $\text{IVA}(\text{OTf}^-)$ to form the six-coordinate intermediates ${}^4\text{IA}(\text{Cl}^-)$ or ${}^6\text{IA}(\text{OTf}^-)$ were computed to be endergonic by 9.5 and 3.7 kcal/mol, respectively. Consequently, their overall barriers for the ODA reaction become much higher (29.9-33.8 kcal/mol via ${}^4\text{TS1A}_{\text{endo}}(\text{Cl}^-)$ and ${}^4\text{TS1A}_{\text{endo}}(\text{OTf}^-)$) than the cationic Fe complex ${}^4\text{IA}$ via ${}^4\text{TS1A}_{\text{endo}}$. These results qualitatively explain no observed reactivity of these two neutral Fe complexes.”

Supplementary Figure 11. Key optimized structures for the formation of **A** for the two neutral Fe(III) complexes in the quartet state in solution by the SMD B3LYP-D3 method. The key distances (Å, in italics), spin density on Fe and relative energies (in kcal/mol at 353.15 K, the sextet state is in the parentheses in purple) by the SMD B3LYP-D3 method are given. Unimportant hydrogen atoms are not shown for clarity.

Supplementary Table 49. The absolute and relative Gibbs free energies (in Hartree) for the key intermediates and transition states with the neutral Fe(III) complexes (see **Supplementary Figure 11**) optimized in solution by the SMD B3LYP-D3 method. (D: doublet; Q: quartet; S: sextet).

		E_{soln}	$(E+ZPE)_{\text{soln}}$	G_{soln}	$G_{\text{soln}-80}^{\circ\text{C}}$	$\Delta G_{\text{soln}-80}^{\circ\text{C}}$
Fe(III)Cl complex						
IVA(Cl)	D	-2712.507668	-2712.22979	-2712.276031	-2712.291746	15.4
	Q	-2712.527632	-2712.250088	-2712.297404	-2712.314112	1.3
	S	-2712.527784	-2712.251840	-2712.300162	-2712.316221	0.0
IA(Cl)	D	-3058.122934	-3057.731720	-3057.789253	-3057.805813	19.1
	Q	-3058.133045	-3057.744082	-3057.803900	-3057.821150	9.5
	S	-3058.130714	-3057.743154	-3057.803329	-3057.820553	9.9
TS1A_{endo}(Cl)	Q	-3292.756029	-3292.222505	-3292.290300	-3292.315960	33.8
	S	-3292.753262	-3292.221261	-3292.288500	-3292.312460	36.0
Fe(III)OTf complex						
IVA(OTf)	D	-3213.725261	-3213.420210	-3213.476301	-3213.493332	15.4
	Q	-3213.747663	-3213.442821	-3213.500565	-3213.517873	0.0
	S	-3213.741920	-3213.438970	-3213.497970	-3213.515619	1.4
IA(OTf)	D	-3559.343687	-3558.924665	-3558.991300	-3559.014920	14.4
	Q	-3559.357785	-3558.941575	-3559.011100	-3559.031730	3.9
	S	-3559.34854	-3558.934337	-3559.003000	-3559.031940	3.7
TS1A_{endo}(OTf)	Q	-3793.984849	-3793.422845	-3793.500200	-3793.523840	29.9
	S	-3793.977665	-3793.417160	-3793.495200	-3793.519010	33.0

As to your comment on the reversibility of the 5-coordinate pathway, we have added the following discussion in the revised manuscript:

“Finally, the six-coordinate mode pathway to form product $^4\text{IIIA}_{\text{endo}}$ is exothermic by 2.6 kcal/mol. Whereas, the five-coordinate mode pathway to form product $^4\text{VIA}_{\text{endo}}$ which is slightly endergonic by 0.4 kcal/mol relative to ^4IA might be a reversible process. Another PhCHO substrate should coordinate to $^4\text{VIA}_{\text{endo}}$ to form a more stable six-coordinate mode product $^4\text{IIIA}_{\text{endo}}$.”

Comment: 1. Fig. 2 contains too much information that is difficult to follow. Authors need to clean this figure and put the less necessary details in Supporting information (e.g. interatomic distances that are not important or mentioned in the discussion in manuscript, and data obtained from other methods). For example, does the molecular geometry shown belong to quartet or sextet Fe species? Differentiation needs to be made and the geometry of

the other Fe species with different spin state can be put in supporting information.

Response: Thanks a lot for your kind and helpful suggestions. We have simplified **Fig. 2** (please see the below revised Figure) and moved some information to SI. As the molecular geometries for the quartet and sextet Fe species are very similar (c.f. key bond length), the molecular geometries of the quartet Fe species was just shown in **Fig. 2**. The key data for the sextet state have been highlighted in orange to be better distinguished from the quartet state.

Revised **Fig. 2.**

Comment: 2. Page 8, line 168, “dispersion attraction was found to lower these reaction barriers” is confusing. What exactly do the authors mean by such statement and what is the origin of this “barrier lowering?”. Does the including of dispersion (D3) change the structure of the reactant? Of the TS? in such way that, in comparison the gas phase structures and geometries, the barrier is lowered? If authors have other information supporting this statement, please provide.

Response: We are grateful for your pointing out these confusions. We used both SMD B3LYP-D3 (with dispersion) and SMD B3LYP (without dispersion) methods for geometry optimization and frequency calculations. Thus, our results and discussion about the dispersion effects includes geometrical and energetic effects. In the revised manuscript, we have revised the below sentence to clarify your comments.

*“Interestingly, dispersion attraction by the SMD B3LYP-D3 method was found to lower these reaction barriers and enhance the driving force of the reaction by roughly ~8-17 kcal/mol, compared to those structures optimized by the SMD B3LYP method (excluding dispersion effect). For instance, the reaction barrier without dispersion via ^{4,6}TS1A_{endo} becomes ~41.1 kcal/mol (see **Supplementary Table 9**).”*

In addition, we have added the below detailed results and discussion about the dispersion effect on the geometry and energies in the revised SI (around **Supplementary Figure 3B**):

*“In contrast to the structures optimized in solution by the SMD B3LYP-D3 (with dispersion) method, the bond between the Fe-O elongated in ^{4,6}IA (by 0.03 Å) and in ^{4,6}IVA (by 0.01-0.03 Å) by the SMD B3LYP (no dispersion) method; the bond between the Fe and the reacting carbonyl O was also elongated in ^{4,6}TS1A_{endo} (by 0.02-0.05 Å) and ^{4,6}TS2A_{endo} (by 0.03-0.04 Å) as well as the bond between the Fe and nonreacting carbonyl O elongated in ^{4,6}TS1A_{endo} (by 0.10-0.12 Å) (see **Supplementary Figure 3B**).”*

*“In addition, the new C-C bond distance in ^{4,6}TS1A_{endo} and ^{4,6}TS2A_{endo} was found to be shorter by 0.02-0.14 Å by the SMD B3LYP (no dispersion) method than that by the SMD B3LYP-D3 (with dispersion) method, whereas the new C-O bond distances were longer by 0.14-0.24 Å. In the uncatalyzed reaction, the new C-C bond distance was not changed, and the new C-O bond distance was elongated by 0.03 Å (see **Supplementary Figure 3B**).”*

Supplementary Table 9. The dispersion effect on the reaction barrier and reaction energy (kcal/mol) to form **A** by the SMD B3LYP-D3 (with dispersion) and SMD B3LYP (no dispersion) methods. The net approximate dispersion effect is given in parentheses. The dispersion effects on the uncatalyzed *exo*-ODA reaction are also presented for comparison. **Notably, the standard state correction was applied here.**

	Reaction Barrier		Reaction Energy	
	With dispersion	No dispersion	With dispersion	No dispersion
Uncatalyzed ODA Reaction				
Endo-type	37.3(-7.8)	45.1	0.5(-7.6)	8.1
Exo-type	38.5(-6.7)	45.2	-2.9(-4.7)	1.8
Fe-Catalyzed endo-type ODA Reaction				
6-c ⁴Fe^a	24.2(-16.9)	41.1	-2.6(-15.4)	12.8
6-c ⁶Fe^a	25.6(-16.2)	41.8	- ^c	- ^c
5-c ⁴Fe^b	24.5(-7.5)	32.0	0.4(-8.8)	9.2
5-c ⁶Fe^b	24.8(-8.3)	33.1	6.1(-7.4)	13.5

a. 6-c stands for the 6-coordinate mode pathway. b. 5-c stands for the 5-coordinate mode pathway. c. Not considered in our calculations.

Supplementary Figure 3B. Optimized key structures for the formation of **A** in the singlet state for the uncatalyzed pathway as well as in the quartet and sextet states (in purple) for the Fe-catalyzed pathways by the SMD B3LYP (no dispersion) method, key distances (Å, in italics) and spin density (s) on Fe are given. Unimportant hydrogen atoms are not shown for clarity.

Comment: 3. Exploring and discussing “The Metal Effect” distracts from the paper. It is more important to focus on the counterion effect since it has a greater impact in reactivity for the same system. In this case, Ru could also be sensitive to solvent, counterion, etc. thus the comparison with Fe seems less impactful and more prone to drawing erroneous computational conclusions.

Response: Thanks a lot for your helpful suggestion. We have moved the “Metal Effect” part to the revised SI and summarized this with “Biological Effect” in only two sentences in the revised manuscript (please see below) after “The Charge Effect of the Fe complexes” section. Also, we have added the results and discussion about the possible counterion effect in “The Charge Effect of the Fe complexes” section in the revised manuscript. Please see our Response to your comment 1.

*“In addition, effects of the ruthenium metal and biological ligand were also examined (see **Supplementary Figures 6** and **Supplementary Table 16**). A considerable Ru(II) character in the ground-state complex $^2\text{IA}_{\text{Ru}}$ and a comparable reactivity using one axial histidine ligand were found.”*

Comment: 4. Exploring the “Biological Effect” section could be moved to the SI or mentioned in the conclusions as potential impact. As it stands, it is very distracting from an already very complex computational and experimental paper.

Response: We are sincerely grateful to your suggestion. We have moved the “The Biological Effect” to SI and summarized this with the “metal effect” in only two sentences in the revised manuscript. Please see our response to your major comment 3.

Comment: 5. Fig. 6 discusses chemoselectivity with Fe complex but in the text the uncatalyzed reaction is also discussed. The authors should also include the uncatalyzed values in this figure and make it more easy to follow.

Response: Thank you so much for your great suggestion. We have revised this Figure in the revised manuscript by including the uncatalyzed values.

Revised **Fig. 6**

Comment: 6. Fig. 8 data from quasi-classical MD simulations is difficult to follow and overall distracts from the paper. Figure 7 is sufficient to highlight the key aspects from these calculations. Authors need to put the less important details (Figure 8) in SI and present the results in a cleaner way in the manuscript.

Response: We would like to sincerely thank your kind comments and suggestion. We have significantly simplified this Figure to only show the time gap of the two bond formation in the gas phase and solution in the revised manuscript. We have moved the remaining results concerning the OEEF to the revised SI (**Supplementary Figure 26**).

Revised **Fig. 7** Distribution in the time gap (in fs) between C-C and C-O bond formation for (a) the uncatalyzed reaction, (b) the 6-coordinate and quartet

state Fe-catalyzed reaction, (c) the 5-coordinate and quartet state Fe-catalyzed reaction, and (d) the 5-coordinate and sextet state Fe-catalyzed reaction in the gas phase (left) and solution (right).

Comment: 7. Fig. 9 contains too much information as a summary figure of the manuscript. It needs to be cleaned. For example, “porphyrin ligand as an auxiliary electron reservoir” can be deleted from the figure.

Response: We appreciate your excellent suggestion. Based on your suggestions, we have simplified and revised Fig. 9 (renamed as Fig.8 in the revised text) as follows:

Revised Fig.

Comment: 8. Figure 7. Please add labels DIRECTLY to the Figure! The authors should not expect readers to read a whole paragraph as the caption to understand the figure!

Response: Thanks a lot for your useful suggestion. We have revised this Figure in the revised manuscript as follows:

Revised Fig.

Comment: 9. Line 166, the authors mentioned “representing two-state mode reactivity”. However, the data supporting this statement is very weak at best since, as shown in Figure 3, the LOWEST energy pathway starts from QUARTET and PROCCEEDS VIA QUARTET. There is no need to make mentioned of such two-state reactivity especially when the data supporting such statement is not strong.

Response: We would like to thank the reviewer for these important comments. In our manuscript, we already mentioned that “several Fe(III) porphyrin complexes with two axial oxygen ligands were experimentally found to have admixed spin states (quartet and sextet states)” (e.g. *Nat. Commun.* **2018**, 4750 and references cited therein). The quartet and sextet states for the related Fe(III) porphyrin complexes are quite similar in energy indeed. In addition, our calculations further showed that the ODA transitions states in the quartet and sextet states become even closer in energy by a few different common DFT methods for 1st row transition metals (see **Supplementary Table 2**). We admit that the highly accurate spin-state energy difference have long been one of the holy grails in computational chemistry. Based on your concerns and the related experimental results, we changed “two-state two-mode reactivity” to “two-mode reactivity with admixed spin states”.

Minor issues:

Comment: 1. Line 25, “a α,β -unsaturated aldehyde” should be “an α,β -unsaturated aldehyde”.

Response: We sincerely thank you for pointing out this correction. We have corrected this.

Comment: 2. Page 4, line 88-89, the solvent used in with SMD model and the corresponding atoms applied with basis set should be mentioned in manuscript instead of only in Supporting Information.

Response: We thank a lot for your helpful suggestion. We have revised the below sentence in the revised manuscript:

“The SMD B3LYP-D3//B3LYP-D3 method with a mixed basis set (6-31G for the C, H, O, N atoms, def2-TZVP for the Fe atom) and a Fe(III)-porphine model catalyst were mainly used”*

Comment: 5. Page 5, line 120, “d5 Fe(III) metal” should be “d⁵ Fe(III) metal”.

Response: Thanks a lot for your careful reading about the format. We have corrected the format/style in the revised manuscript.

Comment: 6. In Fig. 2, authors need to specify that “LUMO” refers to $\pi^*(C=O)$ orbital but not the LUMO of Fe catalytic system (which might partly consist of orbitals of Fe). Further, the authors mentioned the significant LUMO lowering but failed to compare with other LUMO activation models.

Response: We appreciate for your helpful suggestion. We have revised **Fig. 2** to add “ $\pi^*(C=O)$ ” orbital in the revised manuscript. Please see our response to your major comment 1 for the revised **Fig. 2**.

Comment: 7. In Fig.3, do authors have information about the relative energy of product dissociating from Fe center after reaction and the rebound of another aldehyde molecule?

Response: Thank you so much for your suggestion. The relative energy of the dissociation of the product along with re-coordination of another aldehyde molecule to re-generate **⁴IA** is about 0.5 kcal/mol by the SMD B3LYP-D3

method. Such energy is equal to intrinsic reaction energy for the HDA reaction to form **A**. The M06 and particularly M06-2X methods which were developed and tested to give more reliable thermodynamics for main group system (Zhao, Truhlar, *Acc. Chem. Res.* **2008**, 157: "M06-2X has improved performance for main-group thermochemistry, barrier heights, and noncovalent interactions as compared with M06-L and M06") give -6.5~-7.4 kcal/mol. Also, 4 equiv. of the diene substrate with respect to the aldehyde substrate was added, which should promote the ODA reaction. We have added these results and discussion in **Supplementary Table 3B** and **Fig. 3** in the revised SI and manuscript, respectively, as follows:

Supplementary Table 3B. The reaction energy of formation of **A** by different SMD DFT methods in solution. **Notably, the standard state correction was applied here.**

B3LYP-D3	B3PW91-D3	PBE0-D3	M06-D3	M06-2X-D3
0.5	0.3	-11.5	-7.4	-6.5

Revised Fig. 3

“The relative energy of the dissociation of the product **A** along with re-coordination of another aldehyde molecule to re-generate **4IA** is about 0.5 kcal/mol (Fig. 3 and Supplementary Table 3B).”

Comment: 9. In Figure S2 (in SI), illustration of *exo* ODA reaction is missing (neither molecular structure not images). Moreover, labels and energy values are overlapping with one another in Figure S2, making it difficult to follow. Please modify this figure.

Response: We thank the reviewer for this helpful comment and suggestion. We have revised **Supplementary Figure 2** and added images of the *exo*-type molecular structures in **Supplementary Figure 3A** in the revised SI as follows:

Revised **Supplementary Figure 2.**

Revised **Supplementary Figure 3A**. Key optimized *exo*-type structures for the formation of **A** in the quartet state by the B3LYP-D3 method. The key bond lengths (Å, in italics) and spin density on Fe are given. Unimportant hydrogen atoms are not shown for clarity.

Comment: 10. In Fig. 5(a), the authors used comma “,” in label “2,4IA” to indicate different spin states on the left part, while on the right part, “/” was used in label “2TS1Aendo/4TS1Aendo” to differentiate different spin states. In Fig. 5(b) and Fig. 2, comma “,” were usually used. Please modify the labels to stay consistent.

Response: We are also grateful for your helpful comments and suggestions. As you suggested, we have revised them to be fully consistent.

Comment: 11. Page 13, line 279, “effect of using one axial ...” should be “effects of using one axial ...”

Response: Thank a lot for your correction. We have corrected this.

Comment: 12. Page 19, line 409-410, what do “some trajectories” indicate?

More details need to be provided for the statement here to indicate what trajectories show such “intermediate” region and which part of Figure S24 should we look at for these information. Further, in the text, it is mentioned that a total of 1310 trajectories were ran. The authors need to provide the breakdown of these trajectories. How many for solvent? How many for gas phase? How many for uncatat and cat?

Response: We would like to thank you for your helpful comments and concerns. As to the “intermediate” region for “some trajectories”, we have added the below discussion in the revised main text as well as **Supplementary Figure 28** to show those representative trajectories in the revised SI.

“in presence of an OEEF, one out of the 50 productive trajectories for the quartet 5-coordinate mode pathway, and 16 out of the 34 productive trajectories for the sextet 5-coordinate mode pathway do not have the C-O bond formation and stay at the “intermediate” region (where C-C bond is formed, but C-O bond length is still larger than 2.2 Å) after 900 fs propagated from the transition state region”

Supplementary Figure 28. Trajectories that do not form C-O bond after 900 fs for the quartet and sextet 5-coordinate mode pathway in the presence of an oriented external electric field (OEEF).

Also, we have listed the detailed information about all trajectories in **Table 1** and **Supplementary Table 34**. We ran 100 trajectories for each gas phase reaction, except for the quartet 5-coordinate Fe-catalyzed reaction (130 trajectories). In solution, 100 trajectories were ran for uncatalyzed reaction, and 130 trajectories were ran for each Fe-catalyzed reaction in the presence of an OEEF.

Table 1. Number of the DFT quasi-classical trajectories conducted for the uncatalyzed and Fe-catalyzed *endo*-ODA reaction in the gas phase, solution (Soln) and solution in the presence of an OEEF (with a strength of -0.003 au). Percentage (%) of the dynamics stepwise trajectories are given in parenthesis.

	Gas	Soln	Soln + OEEF
Uncatalyzed	100(0)	100(0)	--
6-c ⁴Fe^a	100(21)	130(26)	130(86)
5-c ⁴Fe^b	130(39)	130(54)	130(88)
5-c ⁶Fe^b	100(74)	130(87)	130(96)

^a 6-c stands for the 6-coordinate mode pathway. ^b 5-c stands for the 5-coordinate mode pathway.

Supplementary Table 34. Comparison of the number of the trajectories (N) and the productive trajectories (N_p), average time of the product formation (T_f, fs) and time gap of the two bond formation (T_g, fs) for the uncatalyzed and Fe-catalyzed *endo*-ODA reaction in the gas phase, solution (in parenthesis) and solution in the presence of an OEEF (with strength of -0.003 au, in square brackets) by the B3LYP-D3 and SMD B3LYP-D3 methods.

	N			N _p			T _f (fs)			T _g (fs)		
	Gas	Soln	Soln+ OEEF	Gas	Soln	Soln+ OEEF	Gas	Soln	Soln+ OEEF	Gas	Soln	Soln+ OEEF
Uncatalyzed	100	100	--	97	98	--	32	34	--	5	6	--

d

Fe-Catalyzed Reaction

6-c ⁴ Fe ^a	100	130	130	71	73	63	87	103	220	40	50	174
5-c ⁴ Fe ^b	130	130	130	71	72	50 ^c	123	151	276 ^c	59	98	235 ^c
5-c ⁶ Fe ^b	100	130	130	35	76	34 ^d	182	240	452 ^d	103	199	414 ^d

a. 6-c stands for the 6-coordinate mode pathway. b. 5-c stands for the 5-coordinate mode pathway. c. one trajectory does not form the C-O bond (<1.6 Å) after 900 fs, and thus the time should be longer than this value. d. 16 trajectories do not form the C-O bond (<1.6 Å) after 900 fs, and thus the time should be much longer than this value.

Comment: 13. Page 19, line 423, is “~21-87%” the probability in the absence of an OEEF? Statement needs to be clearer here to make better comparison between probability in the absence and presence of OEEF.

Response: We would like to thank the reviewer for this great suggestion. We have added “in the absence of an OEEF” in the below sentences of the revised manuscript. We have also added the detailed probability in new **Table 1** of the revised manuscript. Please see our response to your minor Comment 12 for **Table 1**.

“~21-87 % in the absence of an OEEF”

Comment: 14. In Fig.9, does “C-C: 1.86-1.93” in 6-coordination mode indicate the distance between two C atoms? Unit needs to be added and clarification should be made on which two carbon atoms are indicated. Also, colors in Fig. 9 are too varied that it looks not clean enough.

Response: We would like to thank you for your kind suggestions. We have revised **Fig. 9** by adding the unit and reduce the colors. Please see our response to your major Comment 7 for this revised Figure.

Comment: 15. Line 177 “were found to be less stable”. This should be clarified. The authors can simply state that it is found higher in energy NOT less stable since “stability” is meaningless. There is kinetic or thermodynamic stability. Which are the authors referring to?

Response: Thanks a lot for your good suggestion. As you suggested, we have revised that part in the revised manuscript as below:

“were found to be higher in free energy than”

Comment: 16. Page 20, line 436-437, the statement “synthetically challenging Fe-catalyzed ODA reaction of the unactivated substrates” looks confusing because ODA reaction is synthetically challenging towards unactivated substrates while in this study Fe catalyst can address this issue. Please modify this statement.

Response: We sincerely thank the reviewer for your suggestions and concerns. We have deleted “synthetically challenging” in that sentence of the revised manuscript to avoid any confusion.

Comment: 17. Figure 4. The authors should include error bars for the experimental KIE.

Response: We appreciate your kind suggestion. The error bars were added in the revised **Fig. 4**. We have added the updated **Fig. 4** in the revised manuscript as follows:

Isotope Effects

(b) Experiment

(c) Isotope effects on the C-H(D) out-of-plane bending

REVIEWERS' COMMENTS:

Reviewer #4 (Remarks to the Author):

Overall, the authors have answer all of my concerns and made the necessary changes to the manuscript to warrant publication in Nature Comm. It is likely that this work will attract broad interest from the chemical community working on catalysis. In particular, the dynamics explored herein are novel and likely will set the foundation for future exploration in the area of organometallic transformations.

I have looked at the details about the 1) critiques from reviewer #3 and the 2) responses by the authors to reviewer #3. Overall, the authors have addressed all issues brought up by reviewer #3 and performed additional calculations in key issues where reviewer #3 was skeptical (the choice of B3LYP in particular). However, there are some comments where the authors could not convincingly address the concerns by Reviewer #3 (in particular, the importance or relevance of quasi-classical dynamics to experiments). That said, I don't think it is possible to address this issue and no matter what the authors will do I doubt that reviewer #3 will see the importance of dynamics. Thus, overall, the authors have addressed all the comments and the paper, with these new changes and additional calculations, is strong and novel for what I expect for Nature Comm.

Response to Reviewers' Comments

We sincerely appreciate the reviewer #4 for his/her useful/constructive comments and suggestions on our revised manuscript. This document summarizes our responses to his/her questions/suggestions and points to the changes we have made in the revised SI, due to limited space and reference in the main text.

The comments from the reviewers (in an italic form)

Our response (in a plain form)

Our revised part in the SI (italic and yellow highlight)

Reviewer #4 (Remarks to the Author):

Comment: *Overall, the authors have answer all of my concerns and made the necessary changes to the manuscript to warrant publication in Nature Comm. It is likely that this work will attract broad interest from the chemical community working on catalysis. In particular, the dynamics explored herein are novel and likely will set the foundation for future exploration in the area of organometallic transformations.*

Response: Thank you so much for your positive comments and support.

Comment: *I have looked at the details about the 1) critiques from reviewer #3 and the 2) responses by the authors to reviewer #3. Overall, the authors have addressed all issues brought up by reviewer #3 and performed additional calculations in key issues where reviewer #3 was skeptical (the choice of B3LYP in particular).*

Response: We sincerely thank your positive comments.

Comment: *However, there are some comments where the authors could not convincingly address the concerns by Reviewer #3 (in particular, the importance or relevance of quasi-classical dynamics to experiments). That said, I don't think it is possible to address this issue and no matter what the authors will do I doubt that reviewer #3 will see the importance of dynamics. Thus, overall, the authors have addressed all the comments and the paper, with these new changes and additional calculations, is strong and novel for what I expect for Nature Comm.*

Response: Thanks a lot for your helpful comments. Based on your suggestions, we have added additional discussion about important results

from the previous dynamics studies in our revised SI (please see the end of “Supplementary Discussion” & “Supplementary References”). We are sorry that we can’t add the below discussion and references in the main text, due to the allowed max. reference numbers (no more than 70).

We added the below discussion in “Supplementary Discussion”

“Compared to the static geometry optimization methods (usually at 0 K), MD simulations with sufficient trajectories include thermal and entropy effects and, thus, should give more realistic ensemble structure. Moreover, MD simulations can give us branching ratio of different products in bifurcate pathways/reactions, while the static geometry optimization cannot. In addition, driven by the thermal effect (and possibly non-equilibrium effect), reaction trajectories do not have to follow minimum energy path (derived from the static geometry optimization) and have shown unusual mechanistic details (e.g. roaming) in some post-transition state dynamics.⁵⁷⁻⁵⁸ Broader entrance channel was observed in our Fe-catalyzed ODA trajectories which should be related to deviate from the minimum energy path.”

We added the below references in “Supplementary References”

57. Sun, L., Song, K. & Hase, W. L. A SN2 Reaction That Avoids Its Deep Potential Energy Minimum. *Science* **296**, 875-878 (2002).

58. Ammal, S. C., Yamataka, H., Aida, M. & Dupuis, M. Dynamics-Driven Reaction Pathway in an Intramolecular Rearrangement. *Science* **299**, 1555-1557 (2003).

59. Townsend, D., Lahankar, S. A., Lee, S. K., Chambreau, S. D., Suits, A. G., Zhang, X., Rheinecker, J.; Harding, L. B. & Bowman, J. M. The Roaming Atom: Straying from the Reaction Path in Formaldehyde Decomposition. *Science* **306**, 1158-1161 (2004).